# Estimating Ground Motion Intensities Using Simulation-Based Estimates of Local Crustal Seismic Response

**Himanshu Agrawal[1] and John McCloskey[1†]**

[1]School of Geosciences, University of Edinburgh, Drummond Street Edinburgh, Edinburgh EH8 9XP, UK

[†]Deceased

Corresponding author: Himanshu Agrawal (himanshu.agrawal@ed.ac.uk), (himansh78@gmail.com)

**Key Points:**

- In the Global South, the absence of seismic catalogues impedes ground motion predictions that are crucial for earthquake-aware urban planning.

- Physics-based simulations can use hypothetical earthquakes to estimate ground motions without extensive earthquake data availability.

- The primary source of short-scale variability in ground motion is the local subsurface geology, making it a crucial focal point.

**Abstract**

It is estimated that 2 billion people will move to cities in the next 30 years, many of which possess high seismic risk, underscoring the importance of reliable hazard assessments. Current ground motion models for these assessments typically rely on an extensive catalogue of events to derive empirical Ground Motion Prediction Equations (GMPEs), which are often unavailable in developing countries. Considering the challenge, we choose an alternative method utilizing physics-based (PB) ground motion simulations, and develop a simplified decomposition of ground motion estimation by considering regional attenuation ($\Delta$) and local site amplification ($A$), thereby exploring how much of the observed variability can be explained solely by wave propagation effects. We deterministically evaluate these parameters in a virtual city named Tomorrowville, located in a 3D layered crustal velocity model containing sedimentary basins, using randomly oriented extended sources. Using these physics-based empirical parameters ($\Delta$ and $A$), we evaluate the intensities, particularly Peak Ground Accelerations (PGA), of hypothetical future earthquakes. The results suggest that the estimation of PGA using the deterministic $\Delta - A$ decomposition exhibits a robust spatial correlation with the PGA obtained from simulations within Tomorrowville. This method exposes an order of magnitude spatial variability in PGA within Tomorrowville, primarily associated with the near surface geology and largely independent of the seismic source. In conclusion, advances in PB simulations and improved crustal structure determination offer the potential to overcome the limitations of earthquake data availability to some extent, enabling prompt evaluation of ground motion intensities.

**Plain Language Summary**

Numerous cities in earthquake-prone regions of the Global South are currently experiencing rapid growth, which poses a significant risk to their populations in the upcoming years. The attainment of effective urban planning, which takes earthquake vulnerabilities into account, typically needs access to long-term earthquake recordings for projecting ground shaking through to future seismic events. Regrettably, the scarcity of earthquake monitoring disproportionately hampers this potential in the Global South, resulting in the utilization of ground motion data from distant locations across the globe. This approach, however, comes with notable limitations

and contributes to the large uncertainty surrounding predictions of ground shaking. We approach this challenge by employing state-of-the-art physics-based simulation techniques that can use hypothetical earthquakes and numerically solve the seismic wave propagation through the Earth's crust. Our study shows that even when a comprehensive earthquake database is lacking, it is feasible to generate reasonably accurate predictions of the spatial variability in expected ground motions using high-resolution local geological information. We emphasize that in cases where urban planning choices need to be formulated for a city characterized by diverse geological features, substantial investments in the measurement of subsurface properties can prove valuable.

## 1 Introduction

The United Nations Human Settlements Programme (UN-Habitat) forecasts that by 2050 some 2 billion new citizens will move to urban centers so that, by then, some 68% of the world's population will live in cities (UN-Habitat, 2022). It is estimated that 95% of this urbanization will happen in the global south. Urban population growth is often accommodated by rapid urban expansion in areas with well-documented seismic risk. The problems of understanding and reducing disaster risk in such rapid development are significant, and while this expansion presents a major global challenge, it also provides a time-limited opportunity to provide evidence-based decision support for this new development (UNISDR, 2015). Efforts in earthquake risk reduction through urban planning guided by high-resolution hazard assessment, could reduce disaster risk for hundreds of millions of these future citizens. This approach also provides a cost-efficient method by concentrating on new constructions, where the expenses related to implementing effective earthquake-resistant design and construction are significantly lower compared to the costs of retrofitting at a later stage.

Seismic hazard analysis informs building codes constraining construction of new development in earthquake prone areas through development of ground motion models (Baker et al., 2021; Bradley, 2019; Kramer, 1996; Kramer & Mitchell, 2006; Mcguire, 2008; Stirling, 2014; Stirling et al., 2012). Observed ground shaking is a result of the interaction between a range of individually heterogeneous fields and processes, leading to deep complexity in even the simplest

relationships. Measures of ground shaking intensity, for example, show an expected systematic
decrease with distance between the observation and source, but the systematics are overprinted
by the interactions between the complexities of the event and the crustal volume explored by the
seismic wave train. The result is high amplitude variability in the observed intensity. Note that
the uncertainty in the observations, in either intensity or distance, makes only a small
contribution to this variability; the variability is an intrinsic part of the process.
Consider a series of events recorded at large number of sensors. In the commonly applied
approach, the analyst chooses a functional form for the systematic decay of intensity and uses
some fitting procedure to estimate its parameters. The resulting model is commonly known as a
Ground Motion Model (GMM) (Douglas & Aochi, 2008; Douglas & Edwards, 2016a, 2016b),
and takes the form:
$$lnIM = \mu_{lnIM} + \sigma_{lnIM}.\epsilon \qquad (1)$$
Where, $IM$ is the required intensity measure, $\mu_{lnIM}$, is the estimated mean-field intensity, $\sigma_{lnIM}$,
is an estimate of the variability around the mean which is usually assumed to conform to a log-
normal distribution and $\epsilon$ is the standard normal variate.
It is important to note that the $\mu_{lnIM}$ term does not just describe the attenuation of intensity with
distance. Common forms of $\mu_{lnIM}$ attempt to parameterize descriptions of the physics of the
entire process including source properties, such as focal mechanism and their resulting
directivity, as well as the local response of the site using estimates of $V_{s30}$ (time-averaged shear-
wave velocity in the top 30m) and $\kappa$ (high frequency attenuation parameter) for example (Aki,
1993; Borcherdt & Glassmoyer, 1992; Bradley, 2011; Hough & Anderson, 1988; Kaklamanos et
al., 2013; Shi & Asimaki, 2017). Expressions for $\mu_{lnIM}$ in current GMMs include numerous
parameters, use advanced statistical techniques to fit these complex functions, and represent a
practical approach to a fundamentally intractable problem (Douglas & Edwards, 2016a).
In practice, an ergodic assumption is invoked in GMM development by aggregating the data
from multiple spatial locations that is assumed to be equivalent to the distribution in time
(Anderson & Brune, 1999). However, with the increasing data for a particular tectonic area, the
non-ergodic or partial non-ergodic approaches are favoured which modify $\mu_{lnIM}$ and $\sigma_{lnIM}$ based
on calibration with the local data that is available (Bradley, 2015; Rodriguez-Marek et al., 2014;
Stewart et al., 2017). It is observed that major component of ground motion amplification can be
associated with the local geological factors e.g. sedimentary basins (Graves et al., 1998; Pilz et
al., 2011; Zhu et al., 2018), surface topography (Lee et al., 2009; Maufroy et al., 2012; G. Wang
et al., 2018), and soil conditions (Bazzurro & Cornell, 2004; Cramer, 2003; Torre et al., 2020).
Hence, the general practice in GMM development is dominated by using near-surface site-
specific parameters (for example $V_{s30}$ and $\kappa$). It is suggested that these near-surface parameters
might exhibit strong correlations with geological features at greater depths, like basin depth
parameters ($Z_{xx}$) (Chiou & Youngs, 2014; Kamai et al., 2016; Tsai et al., 2021), and
consequently the amplification. However, opposing studies show that the amplification patterns
might not necessarily correlate with these parameters (Castellaro et al., 2008; Mucciarelli &
Gallipoli, 2006; Pitilakis et al., 2019), for example, sites with velocity profiles which are not
monotonically increasing with depth. This highlights the necessity to investigate more regional
geological structure to better understand the complexities of ground motion amplification.
Recently, the advances in computational capabilities and understanding the physical processes
have made it possible to use physics-based (PB) simulations for modelling ground motions
(Bradley, 2019; Graves & Pitarka, 2010; Smerzini & Villani, 2012; Taborda et al., 2014). PB
simulations are carried out by numerical modelling of the entire process of rupture
characterization and seismic wave propagation through the potentially complex Earth's crust.
However, the high computational cost and complex input requirements associated with them
restrict the large-scale usage of these methods, particularly in 3D. As a consequence the relative
contribution of these processes to the total observed variability has been relatively unexplored
compared to that of local shallow (decametre) site conditions.
Two immediate problems emerge in enacting the current ground motion modelling approaches in
the context of rapid urbanization in Global South, described above. Firstly, understanding ground
motion requires extensive seismic databases recording appropriate measures of intensity from a
large number of earthquakes, recorded at a network of sensors in the area of interest, for
example, PEER-NGA databases (Ancheta et al., 2014; Atkinson & Boore, 2006; Spudich et al.,
2013). Such catalogues necessitate the deployment of seismometers for many years even in the
most seismically active areas that is not possible to address the current time-critical problem
(Freddi et al., 2021). Secondly, urban development projects require hazard information at
unusually high resolution. Urban flood modelling and landslide susceptibility estimates, for
example, typically strive to use digital terrain models with 2-meter resolution supplemented by
high-resolution geotechnical assessments (Jenkins et al., 2023). Seismic intensity also varies
significantly over the scale of interest for urban planning, particularly where development is
planned over sedimentary basins or near to coasts or rivers with strong spatial contrasts in sub-
surface seismic velocity (Bielak et al., 1999; see also, Cadet et al., 2011; Foti et al., 2019). Some
efforts have been made to incorporate these factors into GMPEs (Abrahamson et al., 2014;
Campbell & Bozorgnia, 2014; Chiou & Youngs, 2014; Marafi et al., 2017), however, the
extensive information required to accurately characterize such effects remains a challenge. As a
result, the potential for high cost-benefit risk reduction that would accrue from high-resolution
understanding of ground motion variability remains elusive. Typically, GMMs developed in
data-rich countries of the global north are reconditioned for deployment in areas for which they
have no obvious physical validity (Hough et al., 2016; Nath & Thingbaijam, 2011). At best, this
leads to poor spatial resolution precluding the detailed site classification that is critical for
seismic microzonation studies needed for cost-effective urban planning (Ansal et al., 2010). The
development of appropriate techniques for rapid, local, high-resolution seismic hazard
assessment is a significant global challenge.
In this research, we approach this challenge by using a simplified decomposition of ground
motions into parametric relations explaining the regional and local variations in the measured
intensity. We focus on the effects only due to the sedimentary basins, which are known to
enhance the amplitude and duration of seismic waves through frequency-dependent focusing,
trapping and resonance (Castellaro & Musinu, 2023; Frankel, 1993; Yomogida & Etgen, 1993).
We demonstrate the usefulness of PB simulations in capturing the primary low frequency (LF),
<1Hz, sedimentary basin effects that contribute to the variation in ground motion within an
*urban* area situated within a seismically active region. We show, to first order, seismic intensity
decays along the wave path according to the integrated rheological properties of the region and is
concurrently subject to relative amplification specific to any point on the surface. We first
provide the theoretical physical basis for the decomposition and then describe the simulation
domain and the numerical scheme used to explore it. We then describe how the main elements of
the problem, i.e., regional mean field attenuation ($\Delta$) and local sie-specific amplification ($A$)
(explained in the subsequent section), can be extracted from the simulations and demonstrate
their use in the reconstruction of originally simulated intensities. We highlight that the
assessment of these reconstructed intensities is not notably influenced by source characteristics
(such as location and directivity). Therefore, calibrating these parameters and understanding
short-scale ground motion amplification variability can address the challenge posed by the lack
of earthquake data. We suggest that this approach, when extended to including Higher
Frequencies (HF), might provide an improved relative seismic risk assessment in the form of
more reliable microzonation maps at the scale of urban planning, which is based on rapid
seismological site characterization in the absence of long duration seismic catalogues.

## 2 Theoretical considerations

Using the seismic representation theorem, (De Hoop, 1958; Knopoff, 1956), in polar coordinates
the displacement $U_{\delta,\varepsilon}$ recorded at a site $\varepsilon$ for a point-source earthquake $\delta$ is given by:

$$U_{\delta,\varepsilon} = G_{\delta(r,\theta,\emptyset),\varepsilon} * f_{\delta(r,\theta,\emptyset)} \qquad (2)$$

Where, $r$ is the distance between source and receiver, and $\theta$ and $\emptyset$ are the positional angles in a
spherical coordinate system, $f_\delta$ is a force vector at $\delta$ and $G$ is the elastodynamic Green's
function providing the displacement at $\varepsilon$ due to $f_\delta$. Since we consider the peak displacement in
elastic medium in what follows, this equation is time invariant.

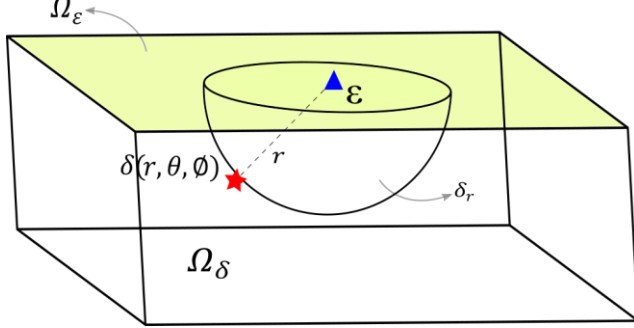


*Figure 1: A cuboidal domain having a receiver at $\varepsilon$ and a seismic point source at $\delta(r,\theta,\emptyset)$. The*
*top surface of this domain represents receiver field $\Omega_\varepsilon$ and the volume defines a source field $\Omega_\delta$.*
*All sources at a distance $r$ from $\varepsilon$ can be represented as the surface of hemisphere $\delta_r$. These*
*ground motion intensity at $\varepsilon$ due to these sources are integrated in equation 3. This can further*
*be integrated for all receivers at the surface $\Omega_\varepsilon$, as calculated in equation 4.*
Consider a receiver at point $\boldsymbol{\varepsilon}$ that experiences displacements due to sources of a given seismic
moment at a point $\boldsymbol{\delta}$ (see Figure 1). The average logarithm of the peak displacement field for all
possible point sources $\boldsymbol{\delta_r}$ at distance $\boldsymbol{r}$ from the receiver $\boldsymbol{\varepsilon}$ can then be expressed as-
$$\overline{\ln\left(U_{\delta_r\varepsilon}\right)} = \frac{1}{2\pi^2}\int_0^\pi\int_0^{2\pi}\ln\left(U_{\delta(r,\theta,\emptyset),\varepsilon}\right)d\theta d\emptyset \tag{3}$$

$\overline{\ln\left(U_{\delta_r\varepsilon}\right)}$ then represents the expectation value for the intensity at $\boldsymbol{\varepsilon}$ due to all possible events at
distance $r$. In this formulation, we consider point sources without any particular focal
mechanism, so equation 3 might be considered as an integration over all possible focal
mechanisms at all possible points on the hemisphere.
Integrating over all receivers $\boldsymbol{\Omega_\varepsilon}$ on the surface of the domain:
$$\overline{\ln\left(U_{(\delta\varepsilon)_r}\right)} = \frac{1}{\Omega_\varepsilon}\iint_{\Omega_\varepsilon}\overline{\ln\left(U_{\delta_r\varepsilon}\right)}d\varepsilon \tag{4}$$

then provides a mean field estimate of the expected intensity for any source-receiver pair
separated by the distance $\boldsymbol{r}$, and a graph of $\overline{\ln\left(U_{(\delta\varepsilon)_r}\right)}$ against $\boldsymbol{r}$, represents the mean field decay
of intensity with distance throughout the entire volume.
The response at a particular location on the surface to any specific event at some distance $\boldsymbol{r}$ will,
of course, be subject to the source, path and site effects, all contributing to some local
modification of the mean field expectation. Consider the ground motion at a receiver $\boldsymbol{\varepsilon}$ due to
any source $\boldsymbol{\delta}$, again, the peak displacement $(\boldsymbol{U_{\delta,\varepsilon}})$ can be calculated using the representation
theorem, this time giving:
$$U_{\delta,\varepsilon} = G_{\delta,\varepsilon} * f_\delta \tag{5}$$

This peak ground displacement $\boldsymbol{U_{\delta,\varepsilon}}$ varies with $\boldsymbol{\varepsilon}$ but from Equation 4, we know its mean across
the surface is $\overline{\ln\left(U_{(\delta\varepsilon)_r}\right)}$ . Normalising the $\boldsymbol{U_{\delta,\varepsilon}}$ by $\overline{\ln\left(U_{(\delta\varepsilon)_r}\right)}$ removes the mean field decay
leading to a normalised displacement $\widehat{U_{\delta,\varepsilon}}$ given by:
$$\widehat{U_{\delta,\varepsilon}} = U_{\delta,\varepsilon} \Big/ \overline{\ln (U_{(\delta\varepsilon)_r})} \tag{6}$$
Finally, to encapsulate the effect of all possible sources at each receiver, this normalised
displacement can be integrated for the entire source field $(\Omega_\delta)$,
giving:
$$\overline{\ln(\widehat{U_\varepsilon})} = \frac{1}{\Omega_\delta} \iiint_{\Omega_\delta} \ln(\widehat{U_{\delta,\varepsilon}}) \, d\delta \tag{7}$$
This $\overline{\ln (\widehat{U_\varepsilon})}$ describes a local normalised amplification expected at any point for all possible
sources. This can be considered as the integrated effect of the whole wave path from all possible
sources that is dominated near $\varepsilon$ where these paths converge. This term introduces the empirical
site-specific variability using the normalised intensity of a suite of earthquakes of any magnitude.
Equations 4 and 7 now allow us to express the final estimate of intensity measure as:
$$\ln(IM) = \overline{\ln (U_{(\delta\varepsilon)_r})} + \overline{\ln(\widehat{U_\varepsilon})} \tag{8}$$
For the sake of simplicity, for an event at $i$, observed at a location $j$, separated by a distance $r$,
$ln\Delta_r$ is used to denote the first term, the mean intensity decay $\overline{\ln (U_{(\delta\varepsilon)_r})}$ and $lnA_j$ defines the
second term describing amplification, $\overline{\ln (\widehat{U_\varepsilon})}$. Now, equation 8 can then be re-written as:
$$IM_{i,j} = \Delta_r * A_j \tag{9}$$
Where $IM_{ij}$ is a non-specific intensity measure recognising that the argument so far may be
generalised to peak velocity or acceleration. $IM_{ij}$ then, provides an estimate of the intensity of
ground motion based on the mean field expected intensity at a distance $\Delta_r$, integrated over the
entire crustal volume under consideration, and a relative amplification $A_j$ due to the integrated
effect of the seismic velocity structure around the site. Both terms on the right hand side are
properties of the crust, regionally and locally, and do not include extended descriptions of the
earthquake source, as we show in the next section. Equation 9 defines the $\Delta - A$ decomposition,
a static ground motion model that emphasises local geology rather than the descriptions of the
earthquake source.
In practice, the mean field $\Delta$ and amplification $\boldsymbol{A}$, can both be calibrated through simulation
based estimates for a given domain, hence the basis is essentially non-ergodic, but it is different
than data-based statistically estimated parameters used in typical non-ergodic GMM (e.g.
Landwehr *et al.*, 2016; Kuehn, Abrahamson and Walling, 2019). The spatial coefficients
estimated in these non-ergodic model are data-dependent, hence in order to find potential drivers
of GM variability in data sparse regions, there is very little scope to use these models. To clarify,
the motivation for the potential utility of $\Delta$-$\boldsymbol{A}$ method is to target the data-sparse regions without
extensive availability of earthquake catalogues.
**3 Defining Domain and source scenarios for simulations**
To explore the behavior and stability of $\Delta$ and $\boldsymbol{A}$ (in equation 9) and how they might be estimated
in practice, we use a virtual world that allows the exploration of the ideas in the absence of
uncertainty but which allows the introduction of precisely constrained variability.  We use a
virtual crustal environment, as shown in Figure 2 (a,b), that incorporates a simplified subsurface
velocity structure centered on a shallow and a deep river basin overlying a crystalline basement
to which simplified velocities have been assigned. The description of the domain includes depth
varying density $(\boldsymbol{\rho})$ , shear wave speed  $(\boldsymbol{V_s})$,  primary wave speed $(\boldsymbol{V_p})$, and anelastic
attenuation factors $(\boldsymbol{Q_p, Q_s})$, and is determined based on the assumed values of these parameters
at the surface of the shallow basin (river channel), deep basin and basement (Brocher, 2005,
2008). The reader is referred to the Jenkins *et al.*, 2023, section 3.1 for detailed description for
crustal domain and earthquake moment distribution. Alternatively, this information is also
accessible in the supplementary materials (Table S1 and Figure S1).
In the middle of crustal domain, we locate a virtual urban environment Tomorrowville (Cremen
et al., 2023; Gentile et al., 2022; Jenkins et al., 2023; Menteşe et al., 2023; C. Wang et al., 2023).
The geology of Tomorrowville is based on a stretch of the Nakhu river valley on the outskirts of
Lalitpur to the south of Kathmandu though the velocity structure described here extends far to
the north and south, and does not represent the actual subsurface seismic velocity in the area.
Instead, we simply generate a hypothetical near-surface velocity structure representative of any
urban settlement located around a river channel set in a deeper and wider sedimentary basin. The
depths of shallow and deep basins in Tomorrowville are presented in Figure 2 (c,d).
The random distribution of 40 thrust-faulting earthquakes (EQ1 to EQ20 are **Mw6** and EQ21 to
EQ40 are **Mw5**) is simulated across the domain (see Figure 2 e,f) using an established physics
based solver, SPEED, which uses Spectral Element Method (SEM) for solving the wave-
propagation equations (Mazzieri, Stupazzini, Guidotti, & Smerzini, 2013; Paolucci et al., 2014;
Smerzini et al., 2011). The SEM combines the geometrical flexibility of the Finite Elements
Method (FEM), i.e., the capability to naturally account for irregular interfaces and mesh
adaptivity, with the high spectral accuracy, i.e., the exponential convergence rate to the exact
solution that results in a fewer number of grid points per wavelength to maintain low dispersion.
The crustal domain has a minimum shear wave velocity of 250 m/s and the smallest element size
of 200m with the spectral degree of 4, hence, the simulations are able to resolve for the
vibrational periods greater than 0.8s. Fault plane dimensions are determined using widely used
empirical relationships developed by Wells & Coppersmith, 1994. Kinematic characterisation of
rupture model is done based on the model developed by Liu et al., 2006; Schmedes et al., 2013 in
which the correlation between the slip, rise time, peak time and rupture velocity among the sub-
faults are derived based on a large ensemble of dynamic rupture simulations of dipping faults.
The moment distribution remains same for each magnitude ensemble, but the strike and dip are
varied. This distribution of rupture scenarios produce a wide range of expected source directivity
for any location. The Peak Ground Acceleration (PGA) maps shown in Figure S2 and Movie S1,
are referred for the visualisation of source orientation and their corresponding effects across the
surface of entire domain. The wavefront evolution for EQ1 can also be found in Movies S2, S3
and S4 of the supplementary information as well.
The Δ-**A** decomposition, developed theoretically above (Section 2), includes no source
variability whereas any attempt to understand seismic hazard must. The azimuth of the events
from the seismometer with respect to the dominant velocity anisotropy introduced by the river
basin will also contribute to the expected ground motion variability. The aim of this manuscript
is not to examine the influence of these features on the observed local intensity; that will follow
in a later work. Instead, we simply explore the extent to which the relative amplification term,
$A_j$, might act as a usable proxy that, to first order, governs the intensity variation across an urban

area, irrespective of the source orientation. This might be considered as a lower bound on the skill of equation 9 in providing the basis for a static site-dependent ground motion model that might be improved later by the introduction of a source term to be constrained by the structural fabric and stress state around any specific location.

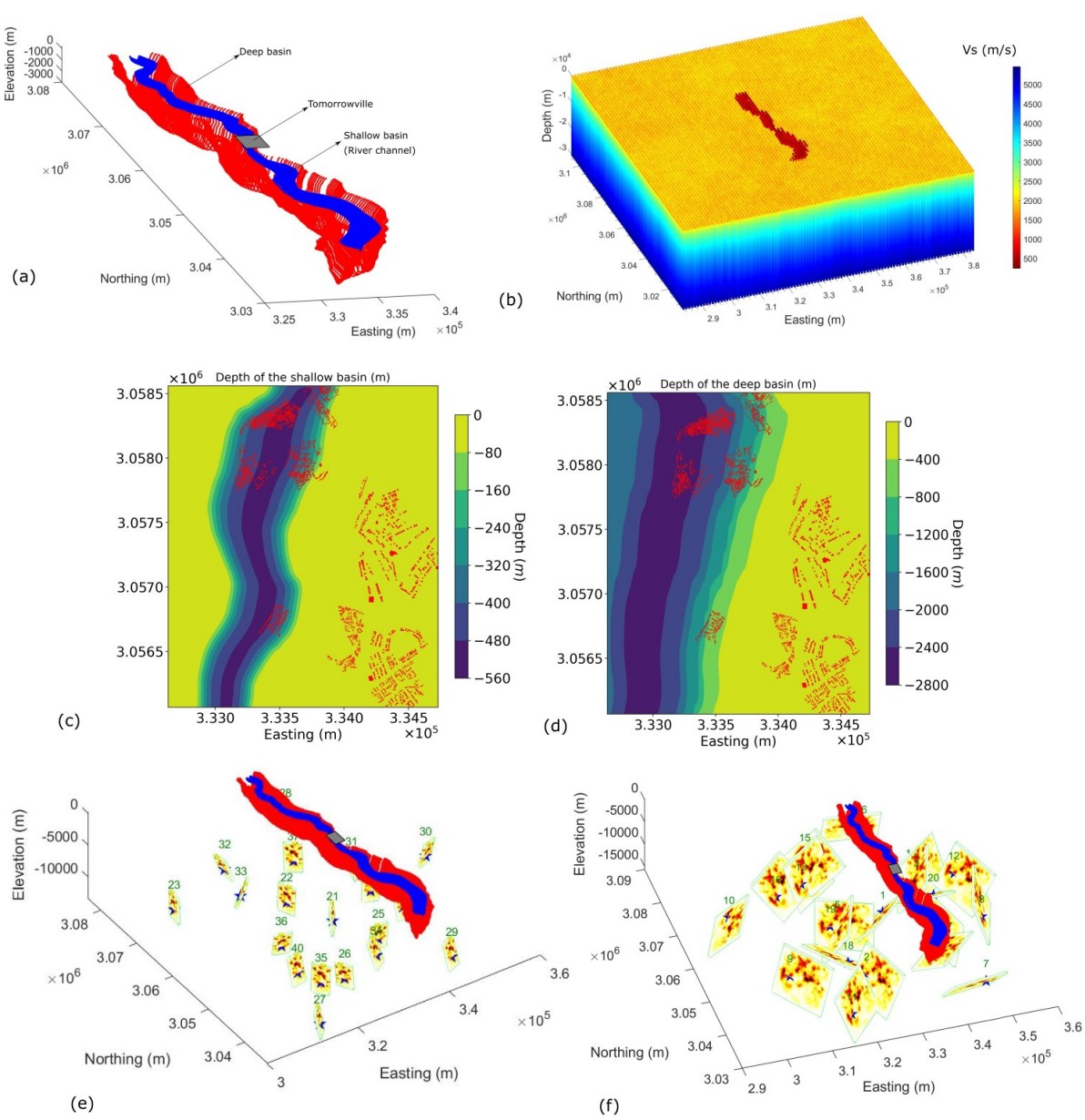

*Figure 2: The computational domain used for the simulations and the distribution of earthquake scenarios is shown. a) The sedimentary basin structure showing a river channel creating a shallow basin of maximum depth 500m located inside a 2km deep basin (see Jenkins et al., 2023*

*for details). The gray rectangle represents Tomorrowville (eg. Cremen et al., 2022, Mentese et*
*al., 2022), which has been designed to help understand the implications of development decision*
*making on consequent risk to future communities. b) Represents the extent of the basin*
*geometries using the shear wave velocities in a crustal volume of dimensions 100 km in length,*
*100km in width and 30km in depth. c) and d) show the basin depths of shallow and deep basins*
*across Tomorrowville with buildings distribution (red polygons). The building distribution is*
*shown to highlight the direct impact of seismicity across the potential future infrastructure. e)*
*and f) show 40 thrust earthquakes with random distributions of dip, rake and strike with EQ21 to*
*EQ40 of **Mw5** and EQ1 to EQ20 of **Mw6** are generated across the domain. The hypocentres*
*are represented by blue stars on the fault surface. The colour distribution across each rupture*
*surface shows the moment release following the kinematic rupture models as developed by Liu et*
*al., 2006; Schmedes et al., 2013.*
**4 Estimation of $\Delta$ and $A$ for Tomorrowville**
The simulation results are used to estimate the $\Delta$ for the crustal domain and $A$ for Tomorrowville
(equation 9). The geometric mean of horizontal components of PGA values are used as intensity
measure for all of the rupture scenarios.
To calculate $\boldsymbol{\Delta}$, we uniformly sample the surface of crustal domain which is a practical and
computationally inexpensive approach to approximate the integration in equation 4.  In the entire
simulation domain, a random set of 100 recording locations is chosen (see green triangles in
Figure 3a) for which estimates of the PGA are simulated for every event, generating a large
number of estimates of the peak amplitude for different epicentral distances giving the data
points for magnitude 5 and 6 events shown in Figure 3b. We use simple least squares regression
to the decay equation:
$$|\Delta_r| = a + b \times ln(r + c) \tag{10}$$
here, $|\Delta_r|$ is an estimation of the mean field intensity measure $\Delta_r$ (introduced in equation 9), $r$ is
the epicentral distance and a,b and c are the empirical parameters evaluated from the data fitting
procedure which might be modified without loss of insight (Figure 3b). The choice of 100
recording locations for $|\Delta_r|$ estimation can have inherent uncertainities based on the selection.
For instance, if the stations are predominantly concentrated in the basin, it could result in higher
intensities in Figure 3b, consequently causing an upward shift in the mean field curve. However,
such a scenario would not uniform sample the entire domain as intended; hence, current choice
of stations seem satisfactory.
It should be noted that the regression method chosen here does not distinguish the repeatable
(within event) and non-repeatable (between events) effects, which is followed from the fact that
each source used here is characteristically similar and is recorded at the exact same set of
receivers. Assuming the entire domain has a homogeneous earthquake distribution, each
recording is considered independent, irrespective of whether the seismic energy is originated
from same or different sources. The concept of earthquake source homogeneity implies that in a
scenario with limited prior knowledge of the tectonics in the area, a reverse faulting earthquake
could potentially occur at any azimuth with respect to the city.

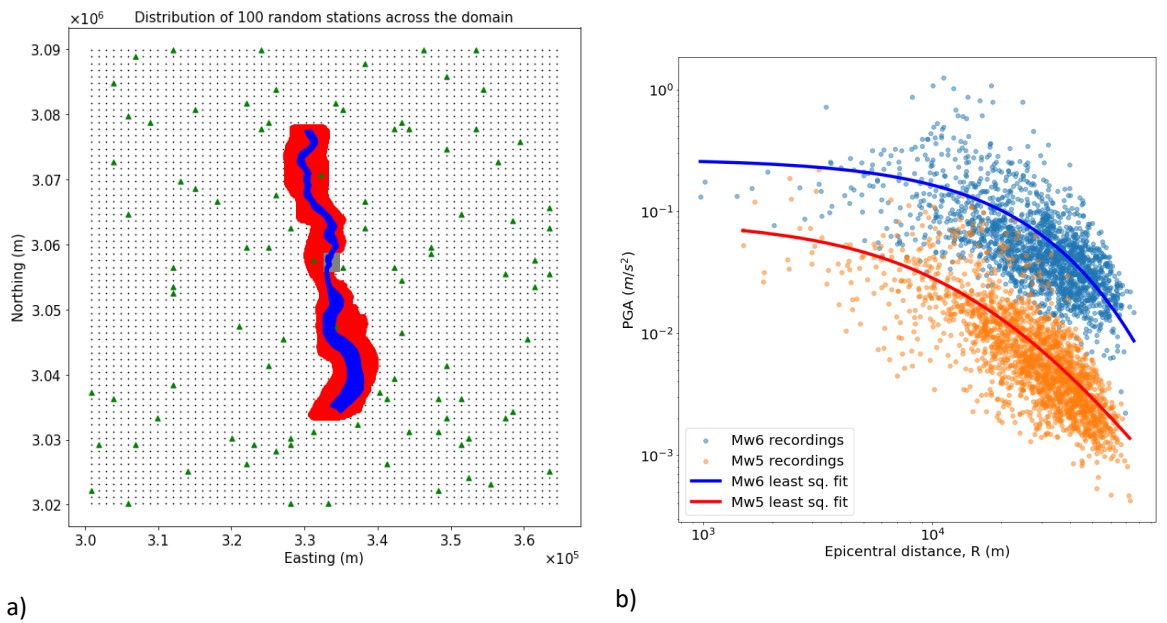

a)          b)

*Figure 3: a) A map of the computational domain showing the shallow basin (blue) created by*
*river channel, and a deep basin (red), as well as the location of Tomorrowville (gray). Green*
*triangles indicate the random locations of the 100 virtual seismometers. b) points indicate PGA*
*versus epicentral distance for each of the 40 events at each virtual seismometer and the curves*
*represents the least squares estimate of the mean field amplitude decay for this data.*
We now must turn our attention to the variability of the data around the curves (Figure 3b) and
will focus on the Tomorrowville sub-domain. Note, any numerical uncertainties due to the
calculation, conditional on the input geological structure, are negligible compared to the
variability observed in Figure 3b. Hence, given the assumption that the simulation is providing
accurate estimates in a virtual setting, each point in Figure 3b accurately represents the local
peak amplitude of waves from a particular event recorded at a single station. To estimate $|A_j|$
for any location $j$, the PGA values from all events are extracted for the Tomorrowville domain
(Figure 4a). Linear interpolation of intensities are used to provide these high-resolution maps,
which sample Tomorrowville at an approximate grid spacing of 28 meters.
As an example, PGA from earthquake 1 (EQ1) is shown along with the spectral accelerations
(5% damped) at 10 stations, S1 to S10 (Figure 4b,c). Please note that these receivers are
positioned within the Tomorrowville domain and are not accounted for in the wider receiver
distribution illustrated in Figure 3a for the evaluation of $|\Delta_r|$. It can be clearly seen that the basin
area is showing strong amplification resulting in higher PGA values due to wave trapping and
resonance of the sedimentary basin layers, as compared to the lower PGA values along the areas
of crystalline basement. Spectral accelerations at 10 stations show different orders of
amplification over the entire period range (0.8s to 5s) corresponding to the geological locations
of these stations. The consistent decrease in amplitude with increasing period observed at all
stations indicates that it is majorly controlled by the selected source spectra. Stations S2, S3 and
S7 lie in the combined (both deep and shallow) basin area and hence, recording maximum
amplification, while the stations S1 and S6 lie above only deep basin area, hence the
amplification is lesser but still significant at higher periods for all three components. The rest of
the stations, S4, S5, S9 and S10 are situated over the basement rocks, hence recording the lowest
value of spectral accelerations.
Our simulations focus on frequencies below 1Hz due to high computational costs associated with
sampling higher frequencies in simulations. However, this analysis remains relevant since basins,
like the Kathmandu basin, often exhibit resonance at similar frequencies (Asimaki et al., 2017;
Oral et al., 2022). Additionally, when dealing with higher frequencies, it becomes necessary to
account for other non-linear site effects that play a significant role in intensity variations
(Semblat et al., 2005), which are not included in this analysis. More discussion on basin
resonance is provided in the supplementary material Text S1.

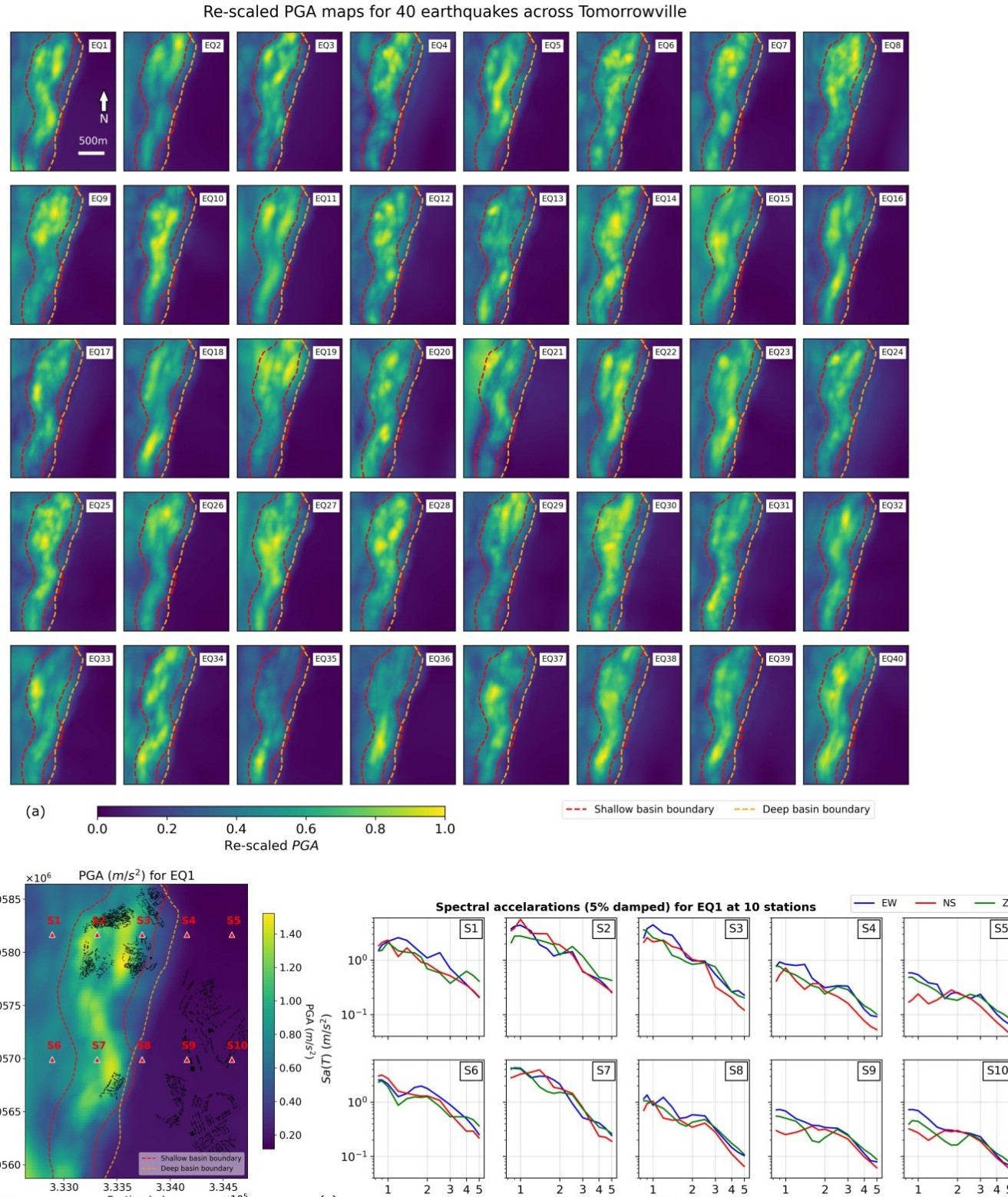

(b)          (c)

*Figure 4: a) PGA maps for 40 events plotted on Tomorrowville city domain. EQ1 to EQ20 represent data from Mw6 earthquakes while EQ21 to EQ40 are for Mw5. Note that we have scaled each map between 0 and 1, where 0 is minimum and 1 is maximum PGA for each earthquake. The similarity of the maps indicates that, to first order, regardless of the absolute value of the PGA across the zone, the relative amplitude for different locations is invariant. b) Shows the PGA (geometric mean of two horizontal components) values for EQ1 along with the boundaries of shallow and deep basins, represented by red and orange dashed lines, respectively. Red triangles show 10 stations, S1 to S10 that are used to show the spectral accelerations for the 0.8s to 5s in c). Three components East-West (EW), North-South (NS) and Vertical (Z) are plotted separately.*

Given the geometry of the basin stretched approximately North-South (NS) whilst being much more confined along East-West (EW), the amplification of both horizontal components should be theoretically contrasting. However, the periods resolved in the simulations show the inter-component variability is still lower than the inter-station variability across different geological domains (Figure 4c). This suggests, the geometric mean of the horizontal components of PGA at each station seem a usable guide to explore the amplification further discussed in this study.

The pattern of higher amplification along the river basin and lower amplification along the basement area is common for PGA maps of all the earthquake scenarios (Figure 4a). Hence while the absolute PGA is strongly dependent on the source magnitude and distance, the *relative* amplitude within any map is qualitatively independent of earthquake source orientation, and even magnitude. The structural similarity of PGA maps in Figure 4a seems to indicate the potential utility of the $\Delta$-$\boldsymbol{A}$ decomposition.

To extract this pervasive feature of relative amplification from all earthquake scenarios we normalise and stack the PGA maps for each event. First, all PGA maps are normalised using the mean smooth earth expectation value $|\Delta_r|$, calculated from equation 10. This normalisation is the practical implementation from the theoretical description given in the equation 6, where the normalisation factor is taken as the mean intensity decay in equation 4. Let, $\left|\boldsymbol{U_{ij}}\right|$ be the

simulated PGA at a particular site $j$ due to an earthquake $i$ at a distance $r,$ then the normalised
PGA $\widehat{|U_{ij}|}$ would be –

$$\widehat{|U_{ij}|} = {|U_{ij}|}\Big/{|\Delta_r|} \tag{11}$$

After normalisation, the average PGA of the normalised maps is calculated for $N_e$ number of
earthquake scenarios, as described in equation 7. This final, averaged PGA map is a
characteristic spatial kernel for the chosen city domain and theoretically contains the average
local amplification ($A_j$) at any site $j$ for any possible earthquake regardless of source, (see Figure
5a). Here, $A_j$ has the following form-

$$A_j = \left( \prod_{i=1}^{N_e} \widehat{|U_{ij}|} \right)^{\frac{1}{N_e}} \tag{12}$$

The calculation of $A_j$ results in a mean amplification field consistent with the spatial variations
observed in the simulations (Figure 5a). Each pixel represents the mean amplification
experienced at that location over all magnitudes, azimuths and directivity.
There is, of course, a dispersion of $ln\widehat{|U_{ij}|}$ values around this mean which is itself a spatially
variable field over the domain, calculated by the $\sigma_{ln\widehat{|U_{ij}|}}$ (Figure 5b) as:

$$\sigma_{ln\widehat{|U_{ij}|}} = \sqrt{\frac{1}{N_e} \sum_{i=1}^{N_e} (ln\widehat{|U_{ij}|} - lnA_j)^2} \tag{13}$$

where, $\sigma_{ln\widehat{|U_{ij}|}}$ gives the variability due to various source scenarios used in the analysis and the
corresponding path effects. The maximum value of $\sigma_{ln\widehat{|U_{ij}|}}$ is 0.56, that is 23.8% of the entire
$lnA_j$ range of 2.35 in Tomorrowville. The difference of 2.35 in maximum ($lnA_{j,max}$) and
minimum ($lnA_{j,min}$) values would mean, the ratio ${A_{j,max}}\Big/{A_{j,min}}$ is $e^{2.35} {\sim} 10.48$, implying an
order of magnitude variation within Tomorrowville. Notably, the ranges of the amplification and
standard deviations are of a realistic order often found in some of the extensively studied real-
world settings as well, for example as shown by Day et al., 2019 in Southern California.
Another approach to understanding the variability of the amplification field involves varying the
number of events used to calculate $lnA_j$ and examining its variability at a specific location using
the events selected through a bootstrapping approach. We chose two stations from Figure 4b, one
representing an area of high amplification over the river basin, named as $S2$, and one in low
amplification over outcropping basement, named as $S9$ (see Figure 5a). The number of events
$N_c$, used to estimate $A_j$, is plotted against the $lnA_j$, where the colour intensity represents the
distribution of the iterations across the entire $lnA_j$ range (Figure 5c). For each $N_c$ value, 100
random combination of events with repetition are used for $lnA_j$ calculation. The red dashes
correspond to the $\pm 1\ \sigma_{s2}$ and $\pm 1\ \sigma_{s9}$ variability around the mean $lnA_j$ value for the respective
$N_c$ value. The convergence of the $lnA_j$ values can be observed even with as low as ~7 events
with a stable $\pm\sigma_{s2}$ and $\pm\sigma_{s9}$ around the $lnA_j$ values of 0.12 each. This distribution of $lnA_j$ is
non-overlapping for both sites, $S2$ and $S9$, which suggests that the local crustal features at both
of these sites is the dominant contributor in the amplification.

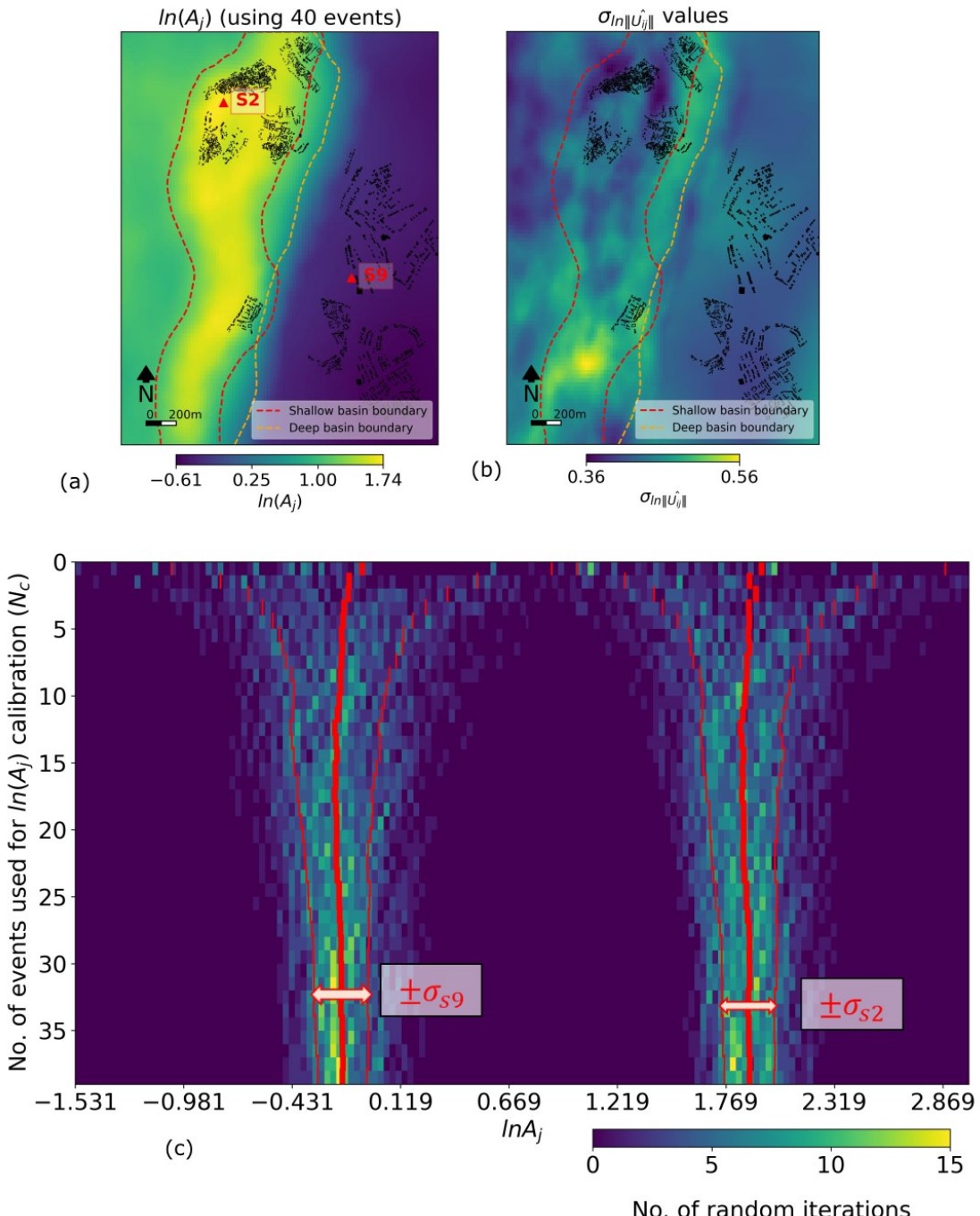

*Figure 5: a) Estimates of $\mathbf{lnA_j}$, and b) the standard deviation ($\boldsymbol{\sigma}_{ln\widehat{|U_{ij}|}}$) for Tomorrowville. Two locations, one in the river basin ( $S2$), and one where the crystalline basement outcrops at the surface at ($S9$) are chosen in a), to plot the convergence of the $\mathbf{lnA_j}$ at $S2$ and $S9$ with an increasing number of events as shown in c).*

## 5 Estimation of PGA using $\Delta$ and $A$ for 40 earthquakes

The theoretical treatment described in section 2 above suggests that the ground motion at a point can be decomposed into the effect of the mean field attenuation over the wave path integrated over the crustal volume and the effect of the local velocity structure. This implies that the reversal of this process should reproduce the original PGA field. Thus if we have robust estimates of $\Delta$ and $A$, then we should be able to reproduce the intensity at any point using equation 9.

We demonstrate this process for a single earthquake, EQ13 located 30.4 km to the NW of Tomorrowville, we will show that the choice of the earthquake is not important. The simulated PGA at every point will be referred to as the true value, $PGA_{true}$ (see Figure 6a,e). To estimate the PGA value explained in equation 9 for this event, referred herein as $PGA_{\Delta A}$, we first calibrate the $\Delta$ (Figure 6b) and $A$ (Figure 6c) using the rest of 39 simulated events. $\Delta$ and $A$ are multiplied as shown in equation 9 to obtain $PGA_{\Delta A}$ values for this earthquake (see Figure 6d). The difference between $PGA_{\Delta A}$ and $PGA_{true}$ is calculated and plotted as a residual map (see Figure 6f). The basin area shows higher negative residuals suggesting underestimation of $PGA_{\Delta A}$ where $PGA_{true}$ values are higher, while surrounding basement exhibits positive values, suggesting overestimation. A graph of $PGA_{\Delta A}$ as a function of $PGA_{true}$ is shown in Figure 6g along with the histograms of all the grid points across Tomorrowville. There is a systematic overestimation of $PGA_{\Delta A}$ values for this particular event at the lower PGA range, and a minor underestimation can be seen at the higher PGA side. This pattern can be attributed to the characteristic that the $lnA_j$ values, which are used to calculate $PGA_{\Delta A}$, have mean amplification values spanning a wider range compared to this specific event. Pearson correlation coefficient ($\gamma$) between logarithms of $PGA_{\Delta A}$ and $PGA_{true}$ is 0.98, suggesting strong correlation between the two. The histograms presented in parallel to the axes also indicate that the distribution nature of PGA remains preserved across Tomorrowville, exhibiting a tri-modal pattern in both $PGA_{true}$ and $PGA_{\Delta A}$ (Figure 6g). This tri-modal pattern is a distinctive influence of three geological domains in the city- the deep basin area (to the left of shallow basin boundary), the area comprising both deep and shallow basins, and the basement region.

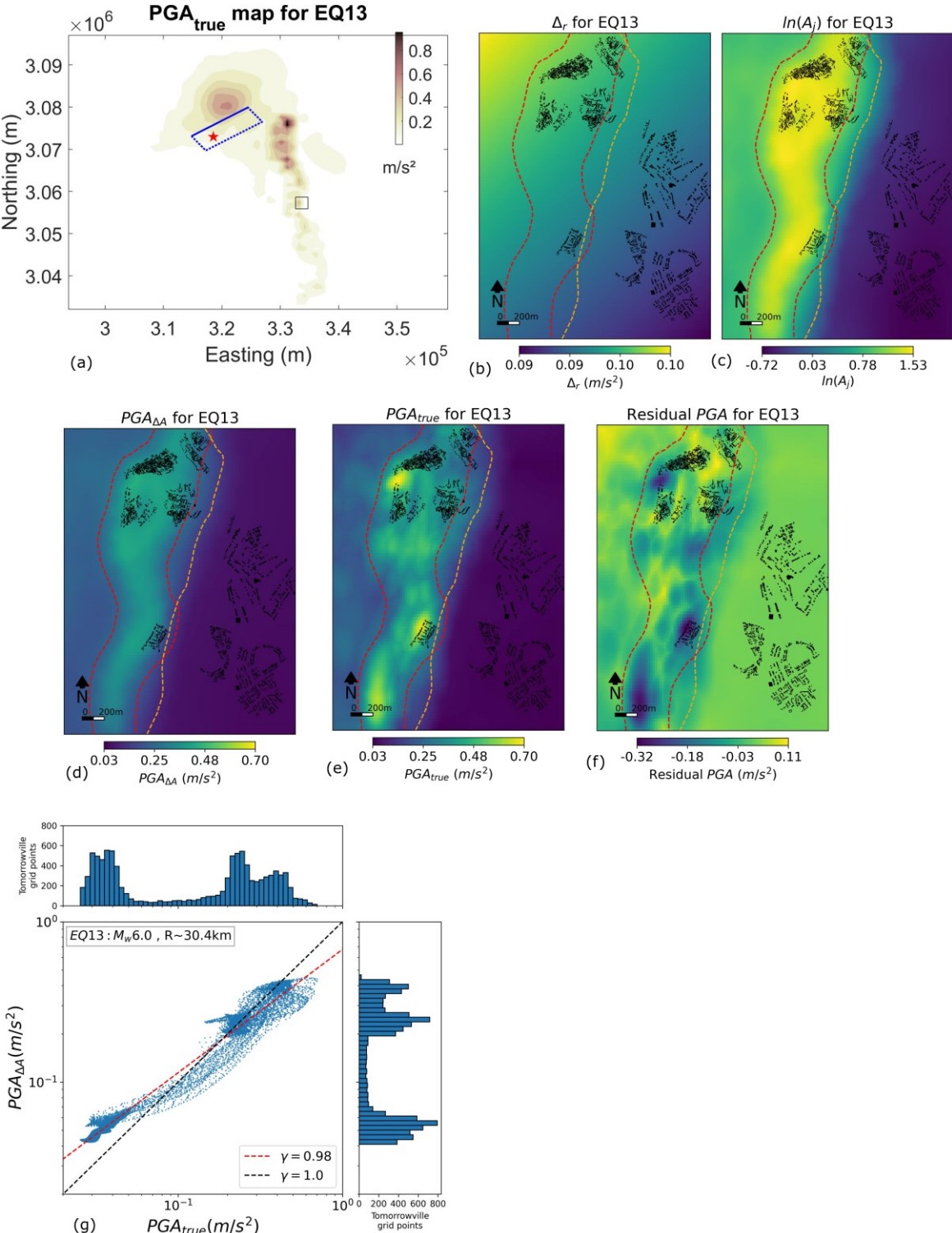


*Figure 6: Result showing estimated parameters for EQ13. a) $PGA_{true}$ map for EQ13 showing*

*the simulation results across the entire crustal domain, the blue dashed-rectangle shows the*

*location of rupture surface (top edge is solid blue), red star shows the hypocentre and black*
*rectangle in the middle of domain shows the location of Tomorrowville. b) shows $\Delta_r$ and c)*
*shows $lnA_j$ for event EQ13 for Tomorrowville. d) shows the $PGA_{\Delta A}$ distribution calculated by*
*multiplying $\Delta_r$ with $A_j$ as conceptualised in equation 9. e) $PGA_{true}$ map for this event obtained*
*through the PB simulation. f) residual between $PGA_{\Delta A}$ and $PGA_{true}$ g) shows the comparison*
*between $PGA_{\Delta A}$ and $PGA_{true}$ for EQ13 using the Pearson correlation coefficient ($\gamma$) of 0.98 for*
*this event. Marginal panels show histograms of $PGA_{\Delta A}$ (right) and $PGA_{true}$(top) indicating the*
*similarity in distribution of $PGA$ values across Tomorrowville city domain.*
Finally, for each event in the suite of 40 earthquakes, the remaining 39 simulations are used to
calculate the $\Delta$ and $A$, that are multiplied to obtain $PGA_{\Delta A}$. The results are compared with the
corresponding $PGA_{true}$ of each earthquake using the $\gamma$ value and best fitting regression line
(Figure 7a). Lowest $\gamma$ value is 0.89, which suggests the correlation is strong for all the
earthquakes. In conclusion, there is a clear potential of predictability in $PGA_{\Delta A}$, with some
variability translated from different source-specific variability due to heterogeneous moment
distribution along the fault surface, as well as, path related variability due to azimuth of sources
with respect to the Tomorrowville. This variability in $PGA_{\Delta A}$, is captured earlier using the
$\sigma_{ln\left|\widehat{U_{ij}}\right|}$ values calculated in Figure 5b.
The impact of source orientation on the obtained $\gamma$ value is illustrated by examining three
parameters: epicentral distance, back azimuth of the earthquake (bearing of the line joining
hypocenter to the center of Tomorrowville), and the angle of approach (the azimuthal difference
between the line connecting the hypocenter to the major fault asperity, and the line connecting
the hypocenter to the center of Tomorrowville) (Figure 7b). The back-azimuth and angle of
approach provide insights into the influence of horizontally anisotropic crustal domain and
directivity effects resulting from variations in fault orientation relative to Tomorrowville,
respectively. $\gamma$ is observed to have a positive trend with epicentral distance indicating that the
earthquakes closer to tomorrowville are poorly constrained by $PGA_{\Delta A}$ compared to the ones
farther away. It can also be seen that the chosen earthquake distribution samples a wide range of
back-azimuth and angle of approach values, indicating a comprehensive representation of these
factors. $\gamma$ does not show any notable trend with the these two factors, hence, their impact on
estimating the distribution of PGA values across Tomorrowville is not substantial.

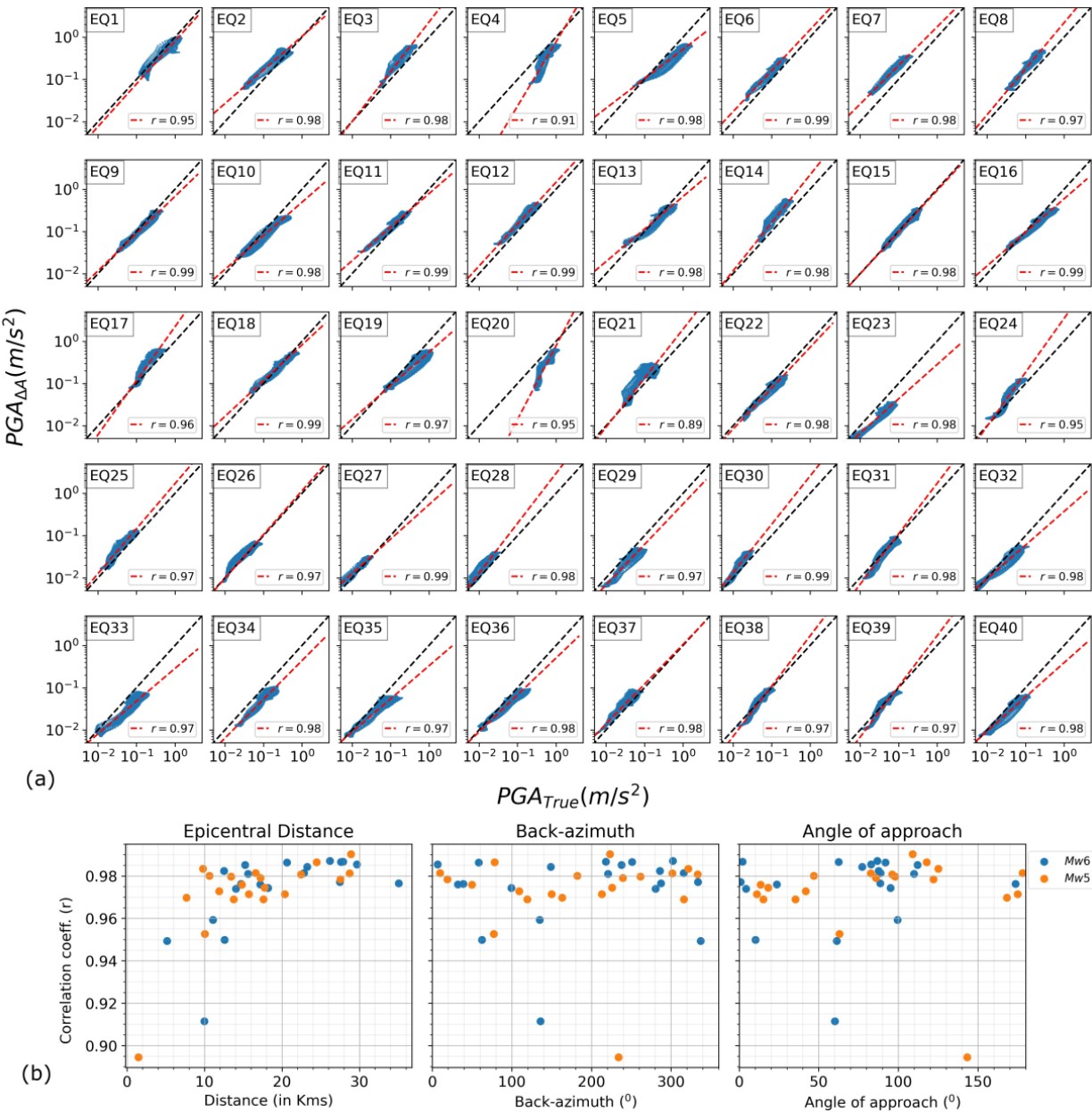

(a)

(b)

*Figure 7: $PGA_{\Delta A}$ is calculated for all 40 earthquakes and compared with the simulated PGA*
*values ($PGA_{true}$). A) Shows the correlation between $PGA_{\Delta A}$ and $PGA_{true}$ for all earthquakes,*
*where red dashed line shows the line of best fit and black dashes show the $\gamma = 1$ line. The $\gamma$*
*value is mentioned for all the earthquakes. B) Shows the $\gamma$ value versus distribution of the*
*following three parameters for all 40 earthquakes- epicentral distance, back-azimuth (bearing of*
*line joining hypocenter to the center of Tomorrowville) and angle of approach (the azimuthal*
*difference between the line connecting the hypocenter to the major fault asperity, and the line*
*connecting the hypocenter to the center of Tomorrowville).*

**6 Discussion and summary**

Estimates from UNDRR suggest that the number of people at risk from a major earthquake will
increase from some 370 million in 2020 to more than 850 million by 2050 (UN-Habitat, 2022).
Due to historically unprecedented rapid urbanization, these people will be increasingly
concentrated in urban centers; the same source estimates that by 2050 global urban population
will increase from the current 56% to around 68% with 95% of this growth happening in the
global south. Without a concerted effort at providing decision support for high cost-benefit risk
sensitive construction, ongoing urbanization in areas of high seismic hazard, will increase
disaster risk for millions.
That the intensity of seismic shaking varies at high spatial frequencies is graphically
demonstrated by large differences of seismic damage over very short distances in areas of
uniform building code (Bielak et al., 1999; see also Asimaki et al., 2012; Dolce et al., 2003;
Ohsumi et al., 2016; Sextos et al., 2018). What is less well known is the extent to which this
variability is the result of differences in the earthquake source, or in contrasts in the rheological
properties of the near surface that might impose a stable and estimable LF amplification, to first
order independent of that source. The former prioritizes forecasting likely earthquake sources in
seismic hazard assessment, while the latter suggests that measuring the properties of the near
surface might produce a pathway to understanding spatial patterns of seismic shaking regardless
of the source. This would in turn open a path to the development of physics-based, high-
resolution building-code classification and support evidence based seismic urban planning
policy.
Current methods for seismic hazard assessment require seismic catalogues built from long-term
deployment of large numbers of seismometers to calibrate ground motion models (Douglas,
2017; Douglas & Aochi, 2008; Douglas & Edwards, 2016a). The observed variability around
these models is assumed to be stochastic and statistical methods are used to provide the moments
of the emerging distributions leading to low spatial resolution estimates of seismic hazard. Over
most of the Global South such long-term data has not been collected nor is there any current
appetite for deploying dense networks of seismometers required for this assessment at the
resolution which would be required to guide seismic risk informed urban planning at actionable
scales.
In this study we have harnessed the potential of high resolution PB earthquake simulations to
explore the extent to which seismic intensity variability might be described by near-surface
geology and that relative seismic intensity is independent of the earthquake source. Do some
areas shake more than others, regardless of the earthquake? We exploit the certainty of a virtual
world, Tomorrowville, in which the rheology, described by the geometry of the seismic velocity,
is known everywhere, in which seismic sources are precisely described by kinematic models
(Graves & Pitarka, 2010; Schmedes et al., 2013), and in which wave propagation is perfectly
described by the wave propagation solver (SPEED) we use (Mazzieri et al., 2013). The choice of
software should not lead to any notable deviation from the results obtained in this study.
The study develops a $\Delta$-$A$ decomposition, that splits the seismic process into a mean-field
attenuation model, describing the amplitude decay with source-receiver distance, and an
amplification field, describing the integrated amplification of the entire wave path as experienced
at each point on the surface. We have shown methods for the estimation of the $\Delta$ model and for
the $A$ field for Tomorrowville and demonstrated that their description can be used estimate the
true PGA field.
This study utilizes PB simulations in a virtual environment that shows a significant fraction of
the observed variability can be explained without categorizing them as stochastic. In the real
world, beyond these deterministic variations, stochastic elements of the process must be
considered separately. Moreover, it becomes important to classify uncertainties as aleatory or
epistemic, when the real data guides the model fitting and resulting deviations (Kiureghian &
Ditlevsen, 2009). However, in this study, PB simulation results are assumed to be devoid of any
modelling uncertainties (or aleatory variability) and they are treated as reproducible true
solutions in the analysis. Consequently, the deviations obtained in the results of Figure 7a are
fundamentally epistemological. The difference between the amplification map for any event and
the $A$ field that determines the value of the local PGA, is precisely quantified and accessible.
Investigations show that the maximum standard deviation of the $A$ field is about 23.8% of the
$lnA_j$ measured across the entire area, that includes the source and path dependent variability.
More importantly, analysis of the variability of the amplification value at any point, indicated
stable convergence from as few as 7 event simulations. Furthermore, comparisons of
amplifications at locations over the river basin with locations on basement in Tomorrowville,
produced stable, order-of-magnitude differences in amplification which converged rapidly and
which gave stable non-overlapping amplification estimates. Of course, both the stability and the
contrast in amplification are functions of the choice of velocity distribution but the choice of
model here was developed to reflect not uncommon velocity geometry not to accentuate
amplification contrasts. We expect that the general conclusions of this work are independent of
the details of the Tomorrowville velocity model.
We have not attempted to explore the variability of the amplification with the source parameters
and the initial results suggest that the influence is not likely to be strong. The main candidates,
source directivity and epicentral azimuth, expected to be dominant in the strongly anisotropic
velocity model used here, do not make an appreciable systematic contribution to the $A$ field.
Descriptions of active fault geometry and seismotectonics of Tomorrowville could impose a
source fabric introducing some systematic influence on the amplification field. Incorporation of
any such influence could only constrain the variability so the results described here might be
considered as a lower bound on the stability of the $A$ field. The primary factor influencing
ground motion amplification in this study is the basin geometry or buried topography, although
the impact of surface topography is also anticipated to significantly affect the amplification
pattern (García-Pérez et al., 2021; Geli et al., 1988; Lee et al., 2009; Poursartip et al., 2020). The
surface topography, often rich in high-resolution data, is the most straightforward to control, and
it is expected to contribute to the observed variability. Future research will concentrate on
investigating the influence of surface topographic features, in addition to buried topography, on
the amplification phenomenon.
The reconstruction of the simulated PGA fields provided further evidence of the efficacy of the
method. Using estimates of the $\Delta$ and $A$ components from a set of 39 simulations provided strong
correlations between true and inverted PGA fields for the 40[th]. Further, in keeping with the
observation of non-overlapping amplification values for basement and basin locations, places
with high shaking were broadly consistently high for all events, locations experiencing low
intensity shaking were also consistent across all events.
The results are suggestive of an underlying physical process in which small-scale LF *relative*
shaking intensity is controlled more by local geology than by source process. Given the
description of the relevant fields through simulations, each taking approximately a day on a
commonly available computer clusters (see Table S3 for simulation parameters and run time
estimates), it is feasible to estimate the entire PGA field ($PGA_{\Delta A}$) for an event of a specific
magnitude and location in milliseconds of computing time. At the minimum, this provides a
workflow through which normal probabilistic seismic hazard assessments, that require estimates
of PGA for thousands of events at each location, can benefit from the advances in physics based
simulations without the massive compute overhead that make these computations unfeasible at
present.
The stability of the relative amplification field together with the stable, order of magnitude
difference in PGA across the surface of Tomorrowville demonstrated in this study, points to
methods for high-resolution seismic hazard estimation based on understanding the static
properties of the near surface, rather than on the unpredictable properties of future earthquakes.
The challenge becomes a problem of measurement, rather than forecasting. There remains the
critical problem either of the elucidation of the velocity structure of the near surface (Sebastiano
et al., 2019), so the $\Delta$ and $A$ fields might be estimated through simulation as in this paper, or the
direct estimation of the field by measurement of the intensity of shaking at high resolution in the
area of interest. To clarify again, this study explores only LF near-surface effects arising from
the presence of complex sedimentary basins and show their contribution in short-scale variability
in amplification. It is noteworthy that these LF effects are additional to the site effects related to
very-near surface (decameter) depths, which include nonlinear soil responses and other high
spatial-frequency velocity variations, all of which can lead to intricate outcomes (Taborda et al.,
2012). Consequently, for applications like enhancing microzonation maps, it's imperative to
merge this analysis with elements accounting for HF variability.
In conclusion, rapid urban expansion in areas of poor historical instrumentation leaves
significant gaps in data for seismic hazard assessment. Furthermore, current methods both
require decade long deployment of dense seismic networks in the area of near-future urban
development and fail to provide high-resolution assessments that identify areas of strong and
weak shaking that could underpin high cost-benefit seismic code classification. The potential of
physics based simulations has prompted the evaluation of the seismic wave field across areas of
near-future development. The results suggest methods to allow the rapid, high-resolution
assessment of geological structure that could lead to risk assessment at unprecedented resolution.

## Statements and Declarations

### Acknowledgments

John McCloskey is listed as a co-author in recognition of his significant contributions.
Unfortunately, he passed away after the manuscript was ready for submission, and we deeply
mourn his loss.
Authors thank initial discussions and simulations obtained with the prompt support and guidance
from Karim Tarbali, former PDRA at the University of Edinburgh. We thank Gemma Cremen,
Chris J. Bean, Mark Naylor, Ian Main, Karen Lythgoe and two anonymous reviewers for
providing constructive feedback and guidance in improving the manuscript.

### Funding

This research is a part of the wider PhD project 'Physics-based Ground Motion Simulations and
Uncertainity Assessment in Rapidly Urbanising Environments'. The PhD student is funded by
University of Edinburgh, School of Geosciences. This research project is also supported by the
Tomorrow Cities Hub (UKRI/GCRF fund under grant NE/S009000/1).

### Author Contributions

Both authors contributed to the study conception and design. Material preparation and data
analysis were performed by HA. The first draft of the manuscript was prepared by HA including
all the figures and text. The text was further reviewed and improved with the help of JM.

### Rights Retention Statement

For the purpose of open access, the author has applied a Creative Commons Attribution (CC BY)
licence to any Author Accepted Manuscript version arising from this submission.
**Open Research**
The data used in this research are mainly the simulation outputs, which are extensive in scale.
The critical information regarding the crustal domain, earthquake hypocenter, and PGA data,
which is pivotal for generating the majority of the manuscript's results, can be found in the
supplementary material. For more detailed information on earthquake moment distribution, we
encourage readers to refer to Jenkins et al. 2023. The software used to run the simulation is an
open-source package, SPEED (Mazzieri et al., 2013). The data analysis and processing is done
using Python and the code is available at https://github.com/himansh78/GroundMotionCalc.git.
**Competing Interests**
The authors declare they have no conflict of interest.

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
