# Peer review of "Crustal Seismic Response"

_EGUsphere, 2023_

## Referee Comment (RC1)

**Broader points**

1) **Introduction:** The motivation for this study is very clear and very timely. The authors stress very well the current gaps in research and how their study will fill these gaps. However, I feel the introduction could be restructured slightly to remove some redundancies. I specifically suggest moving lines 111 – 124 farther up to become the second paragraph of the introduction. That way, the motivation of global urbanization is all at the beginning of this section. Then, I would continue with describing GMPE and their drawbacks and limitations. I would finish this with the very good list of immediate challenges and problems with the current approach (lines 125 – 138 and lines 150 - 157). I would then continue with outlining how you propose to tackle this problem (lines 102 – 110 and lines 139 - 149). The closing paragraph in the Introduction is then lines 158 – 176. After reorganizing the paragraphs, you may find you can shorten some of it to avoid repeating similar arguments.

2) **Section 3:** Please give a brief introduction into the used solver SPEED so the reader can get an idea of the basic concept and its limitations. Is it a finite difference, finite element or finite volume solver? Please list the basic parameters you used such as grid spacing, time increment, and boundary conditions. Please also elaborate why you used this solver. Does it have specific benefits for this study, or could the inclined reader use any wave propagation solver? What features would the solver need to have?

3) **Section 3 (lines 282 – 289):** You state that you do not explore all possible scenarios but will focus on investigating parameter A. That is fine. But then please change your Introduction to reflect this and to not raise false expectations. Specifically, in lines 167 – 168 you state 'we demonstrate the convergence of the simulated ground motions providing measurable fields (Delta and A)' and in lines 170 – 171 you state 'we highlight that the assessment of these parameters is not notably influenced by source characteristics' when in lines 282 – 289 you know tell us that you did not consider 'Delta' in your simulations but will just analyse A. You then proceed in the following sections to discuss both parameters. In essence, be a bit more consistent in wording what you are actually investigating.

4) **Section 4 (lines 312 - 316):** How did you select the random 100 receiver locations? In Figure 3a it looks like only three of the receivers are above the channel and no receiver is actually in Tomorrowville. How will the amount of stations in the basin compared to those outside influence the trends in Figure 3b?

5) **Section 4 (lines 342 - 354)** Receivers shown in Figure 4a are not the same as in Figure 3a. Please elaborate on which receiver geometry you used for which analysis.

6) **Figure 4c:** You show densely interpolated PGA maps. Please give the inter-station distance (and/or receiver layout, see my previous comment) and interpolation method you used to create these images so the reader can understand the degree of smoothing compared to the actual background data that went into these plots.

7) **Section 4 (lines 335 - 337):** You state that numerical uncertainties due to the chosen velocity structure are negligible. Could you elaborate why you draw this conclusion from Figure 3b?

8) **Section 4 (lines 355 - 360):** You mention that high frequencies are often a challenge in numerical simulations. Therefore, you limit your analysis to frequencies below 1 Hz. With your minimal shear wave velocity of 250 m/s, your smallest wavelengths would, thus, be 250m. Most wavelengths will be a lot larger. Could you elaborate on the influence on your interpretations considering the relation between your wavelength and the depth and extent of the basin structure? You hinted at the limitations and benefits from using this frequency range. Could you elaborate a bit more on the significance for hazard assessments for this frequency range considering the spatial dimensions of the channel compared to the wavelength?

9) **Section 6 (Lines 581 - 588):** To convince others to use your new workflow you need to demonstrate that it is either more accurate, requires less input data or is faster. The first two points you tackled quite well in the manuscript. Here, you hint at the speed of the results. To drive the point home, I would be a bit more specific. How long does it take to do all the simulations necessary to calibrate your model? How fast is it afterwards to create the PGA maps? Compare that to the time it takes for the old workflow. The more specific you can be, the more convincing this argument becomes.

10) **Section 6 (Lines 612 – 613):** You did not mention ambient noise tomography at all in your manuscript. Why is this the last sentence? What benefit does this have to your study? If you explain the connection to ambient noise tomography, then move it to its own paragraph in the discussion part. Otherwise, I would suggest removing this last sentence.

11) **Section 6, last paragraph:** I, personally, would prefer to have the last paragraph its own section titled 'Conclusions' and would appreciate if you added 1-2 sentences summarizing your main findings. However, I would be totally ok to leave it like it is.

**Minor Comments**

1) Lines 183 – 184: The Green's function is time invariant when using boundary conditions that are independent in time (i.e., elastic). That is also true for the displacement time series (see for example Aki, K. and Richards, P. (1980). Quantitative Seismology: Theory and Methods). I would rephrase the sentence to say that in the elastic case the Green's function is time invariant.

2) Equation 9: The 'x' is sometimes used to refer to the cross-product of two vectors. Consider changing it to a dot or explaining the symbol in text.

3) Figure 2b: Consider moving you colorbar label from being centered over the plot to being centered over the colorbar.

4) Figure 2b: This is not really necessary to change but it seems that you plot the model domain as individual dots. This creates the effect that the top of the model looks patchy (i.e., one can see the yellow between the orange dots) and there seems to be some streaking that I am not sure if it is a visual glitch or a model feature (i.e., light blue streaks in dark blue at the sides of the model). It might be more representative of the actual model you used if you interpolate the dots to a surface.

5) Supplemental material: While I found the docx document with the supplemental Figures just fine, I was not able to find the referenced movies.

6) Line 315, 337, 338, and all other instances: figure -> Figure
7) Line 426: '... for both sites, S2 and , which ...' There seems to be a missing word
8) Line 453: 'Finally, For each' -> 'Finally, for each ...'
9) Line 469: a 'the' too much

---

## Referee Comment (RC2)

**Major Comments:**

1. In the introduction weak spots of the GMPEs and GMMs in general are mentioned. Site-specific GMPEs may overcome the shortcomings of GMPEs but it requires adequate number of data to work. This can also be added to the Introduction section. Providing an analysis of the GMPEs vs. synthetic PGAs can also give an idea of the how current GMPEs are performing for Tomorrowville.

2. Some of the parameters related to the subsurface features (velocity profile, density etc.) are given in Table S1. It would be better to provide grid resolution in the same table or moving all these information to Section 3.

3. In Line 171 it is mentioned that Δ and A are not influenced by the source characteristics. Since both stations and earthquakes are randomly distributed, there is a chance of some event-station combinations might have directivity effect. As we do not have the exact information related with source characteristics, it is a bit hard to say if the statement in Line 171 is really respected or not.

4. In line 270-273 how kinematics of the fault planes are determined are explained. It would be better to do the same for the fault plane dimensions. Same applies for the rupture velocity summary of the events. These information needs to be presented to better understand the features of the synthetic ruptures.

5. In Line 282-286 it is said that **A** is the parameters that is going to be analyzed. However, in the paper Δ is also densely analyzed.

6. In Line 308, it is said that only the horizontal components of PGA values are used for the intensity measures. However, in Figure 4 vertical components are also included in the spectral acceleration plots. Moreover, non-strike slip faults may produce significant vertical amplitudes. In fact, in Figure 4, in several cases spectral acceleration has larger amplitudes than the horizontal one. Did authors analyzed the amplitudes of the vertical components?

7. In Line 334-337 it is stated that the focus will be on Tomorrowville sub-domain but in Figure 3b the PGA values are retrieved in the whole area not only from the Tomorrowville. Moreover, I do not know how to reach to the conclusion of having negligible uncertainties by looking Figure 3b. Can authors expand this part?

8. Since the station distribution is random, interpolation of PGAs in Figure 4 can be biased. Did author used evenly distributed stations to analyze the effect of station distribution on maps?

9. First three paragraph of the Discussion and Conclusion has a good connection with the beginning of the Introduction section. However, they have nothing to do with the results of the paper. They may reorganized and move to the Introduction section. The subject that is introduced in the Introduction and expanded discussion and conclusion section is a very important topic. It is just not the main topic of the paper. Hence, I believe they need to be presented only in the introduction section.

**Minor Comments:**

1. Line 267 – When the earthquake distribution is introduced the type of faulting also has to be provided. This information is given Line 327.

2. Supplementary Movies cannot be found in Supplementary Material.

3. Figures must be reorganized; subplots has to be better aligned, sizes of the subplots need to be reconsidered.

4. Line 612-613 – I believe this sentence belongs to other parts of the discussion section and needs to be further explained.

5. Some sentences are a bit too long and it can be hard to follow (such as Line 576-579 and Line 589-592).

**Editorial Comments:**

1. Line 81 – "For example" seems unnecessary.
2. Line 179 – significance of the **bold** letters needs to be explained.
3. Line 269 – Citations of the SPEED are given between two dots. The dot in Line 269 should be deleted.
4. Line 276-278 – Movie S4 is mentioned before Movies S1-S3.
5. Line 280 - "… to understand seismic hazard must", I believe it needs to be "is a must".
6. Line 318 – $\Delta_r$ is used two times in the sentence.
7. Line 340-341&343 – Figure 4c is introduced before a and b (Line 343).
8. Super/sub script of some letters are required, eg. Figure 3b y axis label.
9. Figure 6f is neither mentioned nor discussed in the text.
10. Line 379 – "TV" is not introduced.
11. Line 414 – Day et al. 2019 can be cited inside parenthesis in the end of the sentence.
12. Line 426 – The second comma is in the wrong place.
13. Line 545 – Figure 7A should be Figure 7a.
14. Line 566 – "… A field". A should be **bold**.
15. Line 599 – "It's noteworthy …" should be "It is".
16. Bielak and Ghattas 1999 has a doi number. [https://doi.org/10.1061/(ASCE)1090-0241(1999)125:5(413)](https://doi.org/10.1061/(ASCE)1090-0241(1999)125:5(413))
17. Frankel, A. (1993) has a doi number. [https://doi.org/10.1785/BSSA0830041020](https://doi.org/10.1785/BSSA0830041020)
18. Hough and Anderson 1988 has a doi number. [https://doi.org/10.1785/BSSA0780020692](https://doi.org/10.1785/BSSA0780020692)
19. Nath and Thingbaijan 2011 has a doi number. [https://doi.org/10.1007/s10950-010-9224-5](https://doi.org/10.1007/s10950-010-9224-5)

---

## Author Comment (AC3)

Referee comments (RC) are presented in bold text. Following that, Author responses (AR) are given in the regular text, corresponding to each RC. Please note that the line numbers in RCs refer to the previous draft. Line numbers in ARs pertain to the revised draft.

**Referee 1**

**Dear Anonymous reviewer,**

Thank you very much for expressing interest in this work and for your appreciation, as well as for providing valuable comments and suggestions. Your insights have significantly contributed to the enhancement of the manuscript, and I sincerely appreciate the time you dedicated to this manuscript.

Please note that unfortunately, my co-author and dear friend, John McCloskey, passed away last year in November. However, his authorship is retained to honor his contributions to this work. This response file and revisions to the manuscript are now made in consultation with my collaborators (names in the acknowledgement section), who are also experts in seismology.

Kind regards, Himanshu Agrawal

**A. Broader points:**

A1. RC: Introduction: The motivation for this study is very clear and very timely. The authors stress very well the current gaps in research and how their study will fill these gaps. However, I feel the introduction could be restructured slightly to remove some redundancies. I specifically suggest moving lines 111 – 124 farther up to become the second paragraph of the introduction. That way, the motivation of global urbanization is all at the beginning of this section. Then, I would continue with describing GMPE and their drawbacks and limitations. I would finish this with the very good list of immediate challenges and problems with the current approach (lines 125 – 138 and lines 150 - 157). I would then continue with outlining how you propose to tackle this problem (lines 102 – 110 and lines 139 -149). The closing paragraph in the Introduction is then lines 158 – 176. After reorganizing the paragraphs, you may find you can shorten some of it to avoid repeating similar argument

**AR:** The introduction has been reorganised to align with the above recommendations. It now starts by contextualizing urban development in the Global South (lines 57 to 69), followed by an overview of current ground motion modelling approaches (lines 70 to 125), including physics-based methods (lines 117 to 125). Subsequently, potential challenges are outlined (lines 126 to 150), and we then define our approach to addressing these issues (lines 151 to 172).

A2. RC: Section 3: Please give a brief introduction into the used solver SPEED so the reader can get an idea of the basic concept and its limitations. Is it a finite diTerence, finite element or finite volume solver? Please list the basic parameters you used such as grid spacing, time increment, and boundary conditions. Please also elaborate why you used this solver. Does it have specific benefits for this study, or could the inclined reader use any wave propagation solver? What features would the solver need to have?

**AR:** To provide brief introduction to the solver, text has been added from line 266 to 269 to emphasise the unique capability of spectral element solvers to efficiently conduct simulations with high geometrical flexibility and high spectral accuracy. Additionally, lines 552 to 553 in the conclusions

section are incorporated to highlight that the choice of the solver is not important the overall conclusions should remain unchanged even with a different solver.

A3. RC: Section 3 (lines 282 – 289): You state that you do not explore all possible scenarios but will focus on investigating parameter A. That is fine. But then please change your Introduction to reflect this and to not raise false expectations. Specifically, in lines 167 – 168 you state 'we demonstrate the convergence of the simulated ground motions providing measurable fields (Delta and A)' and in lines 170 – 171 you state 'we highlight that the assessment of these parameters is not notably influenced by source characteristics' when in lines 282 – 289 you know tell us that you did not consider 'Delta' in your simulations but will just analyse A. You then proceed in the following sections to discuss both parameters. In essence, be a bit more consistent in wording what you are actually investigating.

**AR:** The phrasing of lines 165 to 167 has been improved to enhance clarity. When we mention- "we highlight that the assessment of these parameters is not notably influenced by source characteristics", instead of 'these parameters', it is now corrected to indicate the intensities that are reconstructed using  $\Delta$  and A. This can also be inferred using Figure 7b, where the correlation of simulated and calculated intensities does not have a defined trend with the source azimuth (location) and angle of approach (directivity).

Furthermore, In the lines 286 to 290, while we mention that we will explore  $A_j$  term mainly, it should not be inferred that  $\Delta$  will be excluded in the rest of the study, rather uncertainties involved in the calculation of  $\Delta$  is not be explored in this work. We are mainly interested in the  $A_j$  field because it has major component of relative amplification within Tomorrowville. Hence, we assume the variability observed in Figure 4a is mainly guided by the  $A_j$  field (Figure 5a).

A4. RC: Section 4 (lines 312 - 316): How did you select the random 100 receiver locations? In Figure 3a it looks like only three of the receivers are above the channel and no receiver is actually in Tomorrowville. How will the amount of stations in the basin compared to those outside influence the trends in Figure 3b?

**AR:** As mentioned in the line 316, the choice of stations is completely random and follows uniform sampling on a 2D surface. The ratio of the area encompassing basin geometries to the total area of the domain of interest is approximately 0.03. Consequently, out of 100 stations, only 3 are deemed to suitably represent the homogeneous distribution across the total surface. The rationale behind choosing the random locations for  $\Delta$  calculation is based on their unimportance in variability observed in high-resolution ground motion intensities (Figure 4a); instead, it is mainly guided by local crustal structure i.e., basin geometry. Hence, the uniform sampling of the entire crustal domain should provide a first order attenuation effect (Figure 6b), which can further be superimposed with the local variabilities of  $A_j$  (Figure 6c), resulting in satisfactory distribution of intensities (Figure 6d) across the local domain of interest. Lines 314 to 315 and 324 to 329 are added to provide the context described above.

Nevertheless, we acknowledge the biases and uncertainties associated with this selection. It has been correctly pointed out that if all stations were located within the basin, it could have led to increased intensities in Figure 3b, potentially resulting in an upward shift in the mean field. However, such a scenario would not uniformly sample the entire domain as we intend. Moreover, in a real-world problem, estimation of  $\Delta$  can be done based on any widely available global GMPEs; for example, CY2014 (Chiou & Youngs, 2014) or BJF93 (Boore et al., 1993) etc. Hence the uncertainties around  $\Delta$  can be problem specific and may not be relevant based on this study. The lines 560 to 569 attempts to provide our approach for uncertainity assessment where we mention that the involvement of real-data would necessitate comprehensive consideration of uncertainties.

A5. RC: Section 4 (lines 342 - 354) Receivers shown in Figure 4a are not the same as in Figure 3a. Please elaborate on which receiver geometry you used for which analysis.

**AR:** The receivers in Figure 4b are used for analysing the local Peak Ground Accelarations (PGA) within Tomorrowville. Conversely, the receivers in Figure 3a are, a completely different set, sampled on a regional scale, which are used for obtaining which is regional crustal attenuation  $\Delta$ . Lines 353 to 355 are added to make this point clear.

A6. RC: Figure 4c: You show densely interpolated PGA maps. Please give the inter-station distance (and/or receiver layout, see my previous comment) and interpolation method you used to create these images so the reader can understand the degree of smoothing compared to the actual background data that went into these plots.

**AR:** The distance between stations within Tomorrowville is 28 meters, indicating that the solver's recordings are made with very high resolution. Consequently, no further smoothing is necessary to generate the depicted figures (Figure 4a, b).

Despite the minimum spectral element size being 200 meters on the surface (as stated in Line 270-271), it is possible to obtain recordings at a much finer resolution using SPEED solver. SPEED achieves this by utilising the Gauss-Lobatto-Legendre (GLL) collocation points within each spectral element. In spectral elements method the solutions are calculated using GLL quadrature (Igel, 2016), where the number of GLL points within a spectral element is decided by the spectral order at which the integrations (for solving wave equation) are performed (In our case, that is 4). Hence, it can be assumed that the integrations are calculated at approximately 50m distance (approximate spacing between GLL points). For assessing the values at recording locations that are 28m apart, the nearest GLL points are identified, and the values are interpolated using the surrounding 64 GLL points with the weights decided based on their distance from the reference node (Mazzieri, 2023). In summary, a linear interpolation scheme is used, that smoothens the intensities obtained through a dense coverage of GLL points, to obtain the intensities at a high resolution. Lines 350 to 351 are added to address the above issue of interpolation.

- A7. RC: Section 4 (lines 335 337): You state that numerical uncertainties due to the chosen velocity structure are negligible. Could you elaborate why you draw this conclusion from Figure 3b?
  AR: The numerical uncertainties conditional on the chosen velocity structure are negligible, which means, assuming the velocity model accurately represents the actual subsurface geology without any uncertainties in a virtual setting around Tomorrowville, the observations at each station recording result from a deterministic outcome of seismic wave propagation (as mentioned in the lines 346 to 348). Even if we assume there aren't any uncertainties in the geological structure, we still recognise there might be uncertainties because of the numerical dispersions in the calculations performed by the solver. However, these uncertainties have a very minor impact and should not impact the overall conclusions derived within this work, as stated in the lines 344 to 346.
- A8. RC: Section 4 (lines 355 360): You mention that high frequencies are often a challenge in numerical simulations. Therefore, you limit your analysis to frequencies below 1 Hz. With your minimal shear wave velocity of 250 m/s, your smallest wavelengths would, thus, be 250m. Most wavelengths will be a lot larger. Could you elaborate on the influence on your interpretations considering the relation between your wavelength and the depth and extent of the basin structure? You hinted at the limitations and benefits from using this frequency range. Could you elaborate a bit more on the significance for hazard assessments for this frequency range considering the spatial dimensions of the channel compared to the wavelength?

**AR:** The frequency range resolved in our simulations is not sufficient to provide an overall picture of hazard assessment, as stated in lines 620 to 625. However, within the resolved range, it is expected that basin-related effects are strongly influencing the peak accelerations that can be observed using spectral accelerations plotted in Figure 4c. It is observed that the basin effects are amplifying the overall spectral content for the stations located in the basin area, i.e., S1, S2, S3, S6 and S7. Now, to understand the extent of this effect, as correctly pointed out, basin dimensions need to be understood.

To understand the 1D basin resonance frequency for fundamental mode ( $f_{1D}$ ), the following relation from Brissaud et al., 2020 can be used:

$$f_{1D} = \frac{V_{s,basin}}{4 * h_{basin}}$$

Where,  $V_{s,basin}$  and  $h_{basin}$  are shear wave velocity and depth of the basin, respectively.

Let us start by analysing the shallow basin, which has a maximum depth of approximately 500m (see Figure 2c in the manuscript), with a minimum shear wave velocity of 250 m/s. Based on these parameters, the expected 1D modal resonance frequency  $(f_{1D})$  can be roughly estimated to be as low as 0.075 Hz. Now, since our simulations are conducted in 3D, we need to consider a suitable approximation for the basin geometry. We can assume that the shallow basin has a predominantly closed curve shape in 2D (East-west and vertical), with the third dimension along North-South being infinite. Therefore, dominant frequency in a 2D model  $(f_{2D})$  will provide a better representation for the 3D basin resonance as compared to  $f_{1D}$ . Given the aspect ratio (depth/width) of the shallow basin, to be approximately 1, it suggests that the  $f_{2D}$  can be estimated to be roughly 2.5 times  $f_{1D}$ (see Figure 16 in Castellaro & Musinu, 2023). This results in  $f_{2D}$  being 0.1875 Hz, which correspond to a 5 second period, approximately. Please note, because the  $V_{s,basin}$  increases with depth (have positive gradient) and the basin's shape being irregular, compounded with the complexity of having two basins in the crustal domain used, the aforementioned equation is just a very generalised representation of basin-resonance. However, the resolved range in our simulations account for the periods up to 5s, we infer that a significant representation of basin resonance and related amplification is included in this analysis (Figure 4c).

As the main objectives of this study do not focus on the specific details of basin resonance, this discussion has been included in the supplementary material (Text S1).

A9. RC: Section 6 (Lines 581 - 588): To convince others to use your new workflow you need to demonstrate that it is either more accurate, requires less input data or is faster. The first two points you tackled quite well in the manuscript. Here, you hint at the speed of the results. To drive the point home, I would be a bit more specific. How long does it take to do all the simulations necessary to calibrate your model? How fast is it afterwards to create the PGA maps? Compare that to the time it takes for the old workflow. The more specific you can be, the more convincing this argument becomes

**AR:** On a machine with 112 cores, simulations typically take around 24 hours to complete, totalling approximately 2688 core-hours. On a typical personal-use laptop machines (with 4 cores), one simulation can take 2688/4=672 hours ~ 28 days. But this duration has been significantly shortened by using High-Performance Computing (HPC) systems with higher number of processors as mentioned above. These simulations are used to estimate the parameters  $\Delta$  and A, and once these values are computed, it only takes milliseconds to calculate the product and generate ground motion intensities for any given scenarios. Table S3 is added to provide estimate for simulation model parameters and run time estimates. Additional lines 603 to 607 are included to indicate the simulation runtime and the potential practicality of the  $\Delta - A$  approach in comparison to the requirements of hazard assessment.

A10. RC: Section 6 (Lines 612 – 613): You did not mention ambient noise tomography at all in your manuscript. Why is this the last sentence? What benefit does this have to your study? If you explain the connection to ambient noise tomography, then move it to its own paragraph in the discussion part. Otherwise, I would suggest removing this last sentence.

**AR:** This study advocates for achieving a detailed understanding of sub-surface characteristics at high resolution, particularly concerning their ability to provide ground motion predictions (as shown through the use of  $A_j$ ). To make this a realistic approach, ambient noise tomography is a most crucial tool as it allows us to provide the high-resolution velocity structure needed to implement the

workflow suggested. We acknowledge the last statement regarding ambient noise tomography might need a little more context than provided, so it has been removed to maintain clarity in the conclusions.

A11.RC: Section 6, last paragraph: I, personally, would prefer to have the last paragraph its own section titled 'Conclusions' and would appreciate if you added 1-2 sentences summarizing your main findings. However, I would be totally ok to leave it like it is.
AR: We believe the conclusions need to be understood within the framework of wider context, which might not be properly conveyed through a separate section. Therefore, we opt to retain them in their current form.

**B. Minor comments:**

- B1. RC: Lines 183 184: The Green's function is time invariant when using boundary conditions that are independent in time (i.e., elastic). That is also true for the displacement time series (see for example Aki, K. and Richards, P. (1980). Quantitative Seismology: Theory and Methods). I would rephrase the sentence to say that in the elastic case the Green's function is time invariant.
   AR: Done
- B2. RC: Equation 9: The 'x' is sometimes used to refer to the cross-product of two vectors. Consider changing it to a dot or explaining the symbol in text.
   AR: Changed to \*
- B3. RC: Figure 2b: Consider moving you colorbar label from being centered over the plot to being centered over the colorbar. ]
   AR: Done
- B4. RC: Figure 2b: This is not really necessary to change but it seems that you plot the model domain as individual dots. This creates the eTect that the top of the model looks patchy (i.e., one can see the yellow between the orange dots) and there seems to be some streaking that I am not sure if it is a visual glitch or a model feature (i.e., light blue streaks in dark blue at the sides of the model). It might be more representative of the actual model you used if you interpolate the dots to a surface. AR: The top of the model is patchy because we have incorporated stochastic variability into the underlying crust by perturbing the velocity at individual points, thereby generating a realistic spatial correlation within the velocity structure. See Jenkins et al., 2023 for more details about the velocity structure generation.
- B5. RC: Supplemental material: While I found the docx document with the supplemental Figures just fine, I was not able to find the referenced movies.AR: We will ask the editor to provide the movies separately.

\_\_\_\_\_

B6. RC: Line 315, 337, 338, and all other instances: figure -> Figure
Line 426: '... for both sites, S2 and , which ...' There seems to be a missing word
Line 453: 'Finally, For each' -> 'Finally, for each ...'
Line 469: a 'the' too much
AR: All fixed.

**Referee 2**

**Dear Anonymous Reviewer,**

*I appreciate the valuable feedback and suggestions you have provided, greatly enhancing the quality of this manuscript. Based on your thorough review, I have attached a detailed response below.*

Please note that unfortunately, my co-author and dear friend, John McCloskey, passed away last year in November. However, his authorship is retained to honor his contributions to this work. This response file and revisions to the manuscript are now made in consultation with my collaborators (names in the acknowledgement section), who are also experts in seismology.

Kind regards, Himanshu Agrawal

**C. Major Comments:**

- C1. RC: In the introduction weak spots of the GMPEs and GMMs in general are mentioned. Sitespecific GMPEs may overcome the shortcomings of GMPEs but it requires adequate number of data to work. This can also be added to the Introduction section. Providing an analysis of the GMPEs vs. synthetic PGAs can also give an idea of the how current GMPEs are performing for Tomorrowville. AR: This point has already been stated in lines (127 to 133), emphasising that the calibration of GMPEs require a larger volume of data.
- **C2.** RC: Some of the parameters related to the subsurface features (velocity profile, density etc.) are given in Table S1. It would be better to provide grid resolution in the same table or moving all these information to Section 3.

**AR:** The data regarding the grid resolution and simulations parameters is now added in supplementary material as a separate table; Table S3.

- C3. RC: In Line 171 it is mentioned that Δ and A are not influenced by the source characteristics. Since both stations and earthquakes are randomly distributed, there is a chance of some event-station combinations might have directivity effect. As we do not have the exact information related with source characteristics, it is a bit hard to say if the statement in Line 171 is really respected or not. AR: Referee 1 also noticed a similar issue in their third comment (see A3). We've now edited that line. For more details, please check the rest of our response to A3 above.
- C4. RC: In line 270-273 how kinematics of the fault planes are determined are explained. It would be better to do the same for the fault plane dimensions. Same applies for the rupture velocity summary of the events. These information needs to be presented to better understand the features of the synthetic ruptures.

**AR:** As lines 273 to 276 state that the kinematic parameterisation is done based on Schmedes et al., 2013 rupture model, which also includes the correlations of other parameters (slip, rise time, peak time) with the rupture velocity. Moreover, since our work doesn't specifically concentrate on source-related parameterization and can equally effectively utilise any other source representation, for example, GP2010 (Graves & Pitarka, 2010) or IM2011 (Irikura & Miyake, 2011), any additional details are not particularly suitable for the main text. However, further details can be found in the supplementary material Figure S1 and S2, where moment distribution across  $M_w 6$  and  $M_w 5$  fault planes are shown. Additionally, hypocentral coordinates for all 40 ruptures are also included as Table S3. Fault plane dimensions are determined using widely used empirical relationships developed by Wells & Coppersmith, 1994. Lines 272 to 273 are added to mention this.

C5. RC: In Line 282-286 it is said that A is the parameters that is going to be analyzed. However, in the paper  $\Delta$  is also densely analyzed.

**AR:** Already addressed in the comment A3.

- C6. RC: In Line 308, it is said that only the horizontal components of PGA values are used for the intensity measures. However, in Figure 4 vertical components are also included in the spectral acceleration plots. Moreover, non-strike slip faults may produce significant vertical amplitudes. In fact, in Figure 4, in several cases spectral acceleration has larger amplitudes than the horizontal one. Did authors analyzed the amplitudes of the vertical components? **AR:** Although we acknowledge the importance of considering the vertical component for non-strikeslip faults, we do not agree with above interpretation of Figure 4c, since both vertical and horizontal components are observed to have similar amplitudes. Due to the complex (non-uniform) shape of the basin geometry (see Figure 2a), which extends predominantly in a North-South direction (Figure 2b), the basin resonance in also expected to be strong on horizontal components and has observed to even exceed the expected higher amplitudes on the vertical component. Nevertheless, we've added the analysis for the vertical component in the supplementary material. The PGAtrue values (derived from the simulations) for the vertical components are depicted in Figure S4. Additionally, the comparison between  $PGA_{\Delta A}$  and  $PGA_{true}$  is provided as Figure S5. Results do not show any significant variation from the results of horizontal component, although slight differences in correlation coefficients can be observed. For instance, event 13, which had a correlation of 0.98 with horizontal component (Figure 7a), has now decreased to 0.95 on vertical component (Figure S5). Event EQ1 has been observed to show an increase in correlation from 0.95 (Figure 7a) to 0.96 (Figure S5). These differences are insufficient to isolate any significant influence of choosing vertical or horizontal components in the analysis.
- C7. RC: In Line 334-337 it is stated that the focus will be on Tomorrowville sub-domain but in Figure 3b the PGA values are retrieved in the whole area not only from the Tomorrowville. Moreover, I do not know how to reach to the conclusion of having negligible uncertainties by looking Figure 3b. Can authors expand this part?

**AR:** In Figure 3b, the  $|\Delta_r|$  is calculated, which represents the regional crustal attenuation or mean field attenuation, and hence, the entire surface of the crustal domain is sampled. In lines 343 to 344, we highlight the variability around the mean field value, as this scatter of points reflects the contribution of local crustal response. It serves as a starting point for understanding the expected high-resolution variability of PGA values across Tomorrowville.

For the second part of the question, we discuss the negligible numerical uncertainties given the chosen velocity structure. This means that, assuming the velocity model accurately represents the actual subsurface geology without uncertainties in a virtual setting around Tomorrowville, the observations at each station recording result from a deterministic outcome of seismic wave propagation (as mentioned in lines 346 to 348). Even if we assume there are no uncertainties in the geological structure, we still acknowledge the possibility of uncertainties due to numerical dispersions in the solver's calculations. However, these uncertainties have a very minor impact and should not affect the overall conclusions drawn within this study, as stated in lines 344 to 346.

C8. RC: Since the station distribution is random, interpolation of PGAs in Figure 4 can be biased. Did author used evenly distributed stations to analyze the effect of station distribution on maps? AR: Station distribution has nothing to do with the Figure 4. Figure 4 shows the simulation results that are created using the PGA values recorded at the unform station distribution across Tomorrowville. The station distribution is only used to calculate the regional crustal attenuation and hence, sampled homogenously. For more discussion around station distribution please refer to A4.

C9. RC: First three paragraph of the Discussion and Conclusion has a good connection with the beginning of the Introduction section. However, they have nothing to do with the results of the paper. They may reorganized and move to the Introduction section. The subject that is introduced in the Introduction and expanded discussion and conclusion section is a very important topic. It is just not the main topic of the paper. Hence, I believe they need to be presented only in the introduction section.

**AR:** The paragraphs at the beginning of discussion section are essential for grasping the complete context of the results and reminding the reader of the overarching motivation. Without these, it would be challenging to connect with the motives behind this work and lose its relevance. The results section serves as a demonstration of the main concept, which aims to highlight the significance of high-resolution variability and its potential application in achieving improved urban planning solutions, particularly in the Global South. However, the technical nature of the results section might lead the reader away from this perspective, making it difficult to arrive at the overall conclusion and potentially getting lost in technical details. Therefore, these paragraphs are retained in current form.

**D. Minor Comments:**

- D1. RC: Line 267 When the earthquake distribution is introduced the type of faulting also has to be provided. This information is given Line 327.
   AR: Added.
- **D2.** RC: Supplementary Movies cannot be found in Supplementary Material. **AR:** Addressed above in B5.
- D3. RC: Figures must be reorganized; subplots has to be better aligned, sizes of the subplots need to be reconsidered.

AR: All figures (except Figure 3) are now resized for better readability.

- D4. RC: Line 612-613 I believe this sentence belongs to other parts of the discussion section and needs to be further explained.
   AR: Already addressed in A10.
- D5. RC: Some sentences are a bit too long and it can be hard to follow (such as Line 576-579 and Line 589-592).

**AR:** Feedback taken but no changes are made.

**E. Editorial Comments:**

E1. RC: Line 81 – "For example" seems unnecessary.

**AR:** Although  $V_{s30}$  and  $\kappa$  are widely used parameters, they are just some examples of the parameters used to account for the local site response, hence "For example" is left as it is. Other types of site response parameters are penetration resistance (N-SPT), undrained shear strength of the upper 30m of ground (Su), and depth to the bedrock etc., that result in similar response characteristics (Chung & Rogers, 2012; McPherson & Hall, 2013; Verdugo, 2019).

- E2. RC: Line 179 significance of the bold letters needs to be explained.AR: Bold letters are only representing equation variables.
- E3. RC: Line 269 Citations of the SPEED are given between two dots. The dot in Line 269 should be deleted.
   AR: Corrected.
- E4. RC: Line 276-278 Movie S4 is mentioned before Movies S1-S3. AR: Order of movies is now rearranged to accommodate this.

- E5. RC: Line 280 "... to understand seismic hazard must", I believe it needs to be "is a must".AR: Original form is right because we intend to say that if our method is used to understand the seismic hazard comprehensively, the source variability must be taken into account.
- E6. RC: Line  $318 \Delta r$  is used two times in the sentence. AR: Corrected.
- E7. RC: Line 340-341&343 Figure 4c is introduced before a and b (Line 343). AR: Figure 4 is now edited with rearranged order.
- E8. RC: Super/sub script of some letters are required, eg. Figure 3b y axis label. AR: Corrected.
- E9. RC: Figure 6f is neither mentioned nor discussed in the text. AR: Lines 456 to 458 are added.
- E10. RC: Line 379 "TV" is not introduced. AR: Replaced TV with Tomorrowville.
- E11.RC. Line 414 Day et al. 2019 can be cited inside parenthesis in the end of the sentence. AR: It is a running text citation hence used without parenthesis.
- E12.RC: Line 426 The second comma is in the wrong place. Line 545 – Figure 7A should be Figure 7a. Line 566 – "... A field". A should be bold. Line 599 – "It's noteworthy ..." should be "It is". AR: Fixed

E13.RC: Bielak and Ghattas 1999 has a doi number. https://doi.org/10.1061/(ASCE)1090-0241(1999)125:5(413) Frankel, A. (1993) has a doi number. https://doi.org/10.1785/BSSA0830041020 Hough and Anderson 1988 has a doi number. https://doi.org/10.1785/BSSA0780020692 Nath and Thingbaijan 2011 has a doi number. https://doi.org/10.1007/s10950-010-9224-5 AR: All added

**References**

- Boore, D. M., Joyner, W. B., & Fumal, T. E. (1993). Estimation of response spectra and peak accelarations from Western North American earthquakes: An Interim Report. In USGS Open-File Report.
- Brissaud, Q., Bowden, D. C., & Tsai, V. C. (2020). Extension of the Basin Rayleigh-Wave Amplification Theory to Include Basin-Edge Effects. *Bulletin of the Seismological Society of America*, 110(3), 1305–1322. https://doi.org/10.1785/0120190161
- Castellaro, S., & Musinu, G. (2023). Resonance versus Shape of Sedimentary Basins. *Bulletin of the Seismological Society of America*, 113(2), 745–761. https://doi.org/10.1785/0120210277
- Chiou, B. S. J., & Youngs, R. R. (2014). Update of the Chiou and Youngs NGA model for the average horizontal component of peak ground motion and response spectra. *Earthquake Spectra*, *30*(3), 1117–1153. https://doi.org/10.1193/072813EQS219M
- Chung, J. won, & Rogers, J. D. (2012). Seismic site classifications for the St. Louis urban area. *Bulletin* of the Seismological Society of America, 102(3), 980–990. https://doi.org/10.1785/0120110275

- Graves, R. W., & Pitarka, A. (2010). Broadband ground-motion simulation using a hybrid approach. Bulletin of the Seismological Society of America, 100(5 A), 2095–2123. https://doi.org/10.1785/0120100057
- Igel, H. (2016). The Spectral-Element Method. In H. Igel (Ed.), *Computational Seismology: A Practical Introduction*. Oxford University Press. https://doi.org/10.1093/acprof:oso/9780198717409.003.0007
- Irikura, K., & Miyake, H. (2011). Recipe for predicting strong ground motion from crustal earthquake scenarios. *Pure and Applied Geophysics*, *168*(1–2), 85–104. https://doi.org/10.1007/s00024-010-0150-9
- Jenkins, L. T., Creed, M. J., Tarbali, K., Muthusamy, M., Trogrlić, R. Š., Phillips, J. C., ... McCloskey, J. (2023). Physics-based simulations of multiple natural hazards for risk-sensitive planning and decision-making in expanding urban regions. *International Journal of Disaster Risk Reduction*, 84, 103338. https://doi.org/https://doi.org/10.1016/j.ijdrr.2022.103338
- Mazzieri, I. (2023). SPEED source code. https://speed.mox.polimi.it/DOWNLOAD/manual\_2023/html/GET\_\_MONITOR\_\_VALUE\_8f90.h tml
- McPherson, A., & Hall, L. (2013). Site classification for earthquake hazard and risk assessment in Australia. *Bulletin of the Seismological Society of America*, *103*(2 A), 1085–1102. https://doi.org/10.1785/0120120142
- Schmedes, J., Archuleta, R. J., & Lavallee, D. (2013). A kinematic rupture model generator incorporating spatial interdependency of earthquake source parameters. *Geophysical Journal International*, *192*(3), 1116–1131. https://doi.org/10.1093/gji/ggs021
- Verdugo, R. (2019). Seismic site classification. *Soil Dynamics and Earthquake Engineering*, 124, 317–329. https://doi.org/10.1016/J.SOILDYN.2018.04.045
- Wells, D. L., & Coppersmith, K. J. (1994). New empirical relationships among magnitude, rupture length, rupture width, rupture area, and surface displacement. *Bulletin - Seismological Society* of America, 84(4), 974–1002. https://doi.org/10.1785/bssa0840040974